# Structural and mechanistic analysis of a tripartite ATP-independent periplasmic TRAP transporter

Martin F. Peter [1], Jan A. Ruland [2], Peer Depping [1,3], Niels Schneberger[1], Emmanuele Severi [4,5], Jonas Moecking [1], Karl Gatterdam[1], Sarah Tindall[4], Alexandre Durand [6], Veronika Heinz [7], Jan Peter Siebrasse[2], Paul-Albert Koenig [8], Matthias Geyer [1], Christine Ziegler [7], Ulrich Kubitscheck [2], Gavin H. Thomas [4] & Gregor Hagelueken [1]✉

Tripartite ATP-independent periplasmic (TRAP) transporters are found widely in bacteria and archaea and consist of three structural domains, a soluble substrate-binding protein (P-domain), and two transmembrane domains (Q- and M-domains). HiSiaPQM and its homologs are TRAP transporters for sialic acid and are essential for host colonization by pathogenic bacteria. Here, we reconstitute HiSiaQM into lipid nanodiscs and use cryo-EM to reveal the structure of a TRAP transporter. It is composed of 16 transmembrane helices that are unexpectedly structurally related to multimeric elevator-type transporters. The idiosyncratic Q-domain of TRAP transporters enables the formation of a monomeric elevator architecture. A model of the tripartite PQM complex is experimentally validated and reveals the coupling of the substrate-binding protein to the transporter domains. We use single-molecule total internal reflection fluorescence (TIRF) microscopy in solid-supported lipid bilayers and surface plasmon resonance to study the formation of the tripartite complex and to investigate the impact of interface mutants. Furthermore, we characterize high-affinity single variable domains on heavy chain (VHH) antibodies that bind to the periplasmic side of HiSiaQM and inhibit sialic acid uptake, providing insight into how TRAP transporter function might be inhibited in vivo.

[1] Institute of Structural Biology, University of Bonn, Venusberg-Campus 1, 53127 Bonn, Germany. [2] Institute for Physical und Theoretical Chemistry, University of Bonn, Wegelerstr. 12, 53127 Bonn, Germany. [3] Aston Centre for Membrane Proteins and Lipids Research, Aston St., B4 7ET Birmingham, UK. [4] Department of Biology (Area 10), University of York, York YO10 5YW, UK. [5] Biosciences Institute, Newcastle University, Newcastle NE2 4HH, UK. [6] Institut de Génétique et de Biologie Molecule et Cellulaire, 1 Rue Laurent Fries, 67404 Illkirch Cedex, France. [7] Institute of Biophysics and Biophysical Chemistry, University of Regensburg, Universitätsstr. 31, 93053 Regensburg, Germany. [8] Core Facility Nanobodies, University of Bonn, Venusberg-Campus 1, 53127 Bonn, Germany. ✉email: hagelueken@uni-bonn.de

Tripartite ATP-independent periplasmic (TRAP) transporters represent a structural- and functional mix of the well-studied ATP-binding cassette (ABC) transporters and secondary active transporters, by functioning as substrate-binding protein (SBP) dependent secondary transporters[1–4]. They are widespread in bacteria and archaea, especially in marine environments, but absent in eukaryotic organisms. TRAP transporters, together with ABC importers and tripartite tricarboxylate transporters (TTT)[5,6], define the three classes of SBP-dependent transporters. In addition to a high-affinity SBP (also named the P-domain for TRAP transporters) that freely roams the periplasm in Gram-negative bacteria, TRAP transporters consist of a smaller- (Q) and a larger- (M) membrane domain. The latter two domains are either fused into a single polypeptide chain or expressed as separate proteins that form a tight complex[7,8]. The best-studied TRAP transporters are HiSiaPQM and VcSiaPQM, two sialic acid (N-acetylneuraminic acid) transporters from the important bacterial pathogens *Haemophilus influenzae* and *Vibrio cholerae* that are the causative agents of meningitis and cholera, respectively[9–11]. For both pathogens, sialic acid uptake by TRAP transporters is essential for virulence and host colonization[10,12–15] and could provide novel targets for the development of antimicrobials, for which the World Health Organization (WHO) has identified ampicillin-resistant *H. influenzae* as a priority pathogen[16].

Of the three classes of SBP-dependent transporters, high-resolution structures for all domains are only available for ABC importers[17]. TRAP transporters and TTTs have so far proved recalcitrant to experimental attempts at elucidating their structures. To date, only structures of the soluble P-domains, mainly from the family of DctP-like SBPs[18–24], have been determined. Features of the SBPs, such as the role of a conserved arginine residue for high-affinity substrate binding and specificity, and the conformational rearrangement of the protein upon substrate binding have been characterized in detail[25–27].

In this work, we determine the structure of the 70 kDa membrane domains of the HiSiaPQM TRAP transporter in lipid nanodiscs with cryo-EM. An essential step for the successful 3D-reconstruction is the generation of a HiSiaQM-specific single variable domain on heavy chain (VHH) antibody and ultimately its extension to a "megabody"[28] that allows us to identify HiSiaQM-filled nanodiscs during 2D classifications and aids in particle alignment. Structural analysis of the TRAP transporter structure reveals a similarity to VcINDY, a dimeric elevator-type[29] dicarboxylate transporter from *V. cholerae*[30–32]. Further, we use in silico modelling, mutagenesis, and in vivo experiments to postulate a model for the tripartite complex, where the known conformational changes of the SBP are coupled to the elevator movement of the transmembrane domains. We use single-molecule tracking total internal reflection fluorescence (TIRF) microscopy and surface plasmon resonance (SPR) to analyze the formation of the tripartite complex and to investigate the effect of interface mutants on complex formation. In addition, we show that VHHs, which specifically bind the QM-domains, inhibit the TRAP transporter function in vivo. This observation, in combination with the proposed transport mechanism, represents an important step toward developing antimicrobial compounds that block TRAP transporters and by this, prohibit the uptake of important metabolites in pathogenic bacteria.

## Results

**Cryo-EM structure of a TRAP transporter**. We expressed the 70 kDa HiSiaQM transporter in *Escherichia coli* MC1061 cells and purified the dodecyl-β-D-maltoside (DDM) solubilized protein to homogeneity (Supplementary Fig. 1). For cryo-EM experiments, we reconstituted HiSiaQM in MSP1D1-H5-bound DMPC lipid-nanodiscs[33] where helix 5 of the MSP1D1 was deleted to create nanodiscs of smaller diameter with 75% of the size of native MSP1D1 nanodiscs[34]. We further created a HiSiaQM-specific megabody[28] (Mb3) to "mark" the location of the transporter inside the nanodiscs (Supplementary Fig. 2). In addition, Mb3 was used during protein sample preparation to pull down the HiSiaQM-filled nanodiscs and thereby to remove empty nanodiscs. The 3D-reconstruction was performed with a combination of cryoSPARC and RELION[35,36] and the resulting volume had a gold standard FSC resolution of 4.7 Å with central parts of the protein resolved up to 3.7 Å local resolution (Supplementary Fig. 3 and Supplementary Table 1). Clear and consecutive density for the polypeptide chain was observed, with many visible side chains, allowing us to confidently model both the transporter and the VHH$_{QM}$3 portion of Mb3, including its complementarity determining regions (CDRs) (Fig. 1 and Supplementary Figs. 4, 5). The helical register of the model is consistent with both the observed sidechains and an AlphaFold prediction (Supplementary Fig. 6)[37]. The correct tracing of the chain was verified experimentally as detailed in the following sections. The position of the bound Mb3 (Fig. 1), unequivocally identifies the periplasmic side of the transporter (see below). We compared our model to a predicted model of the related YiaMN TRAP transporter from Ovchinnikov et al. (~33% amino acid identity to HiSiaQM)[38]. The two structures superimpose with an r.m.s.d. of 5.8 Å over 531 C-alpha atoms, showing that the overall fold of the TRAP transporter was clearly correctly predicted at the time (Supplementary Fig. 7).

HiSiaQM consists of 15 TM helices (Q1–Q4 and TM1–11) and two helical hairpins (HP1 and HP2), which do not cross the lipid bilayer (Fig. 1). Residues 1–149 (TM helices Q1–Q4) form the Q-domain and these four long TM helices are inclined at a 45° angle against the membrane normal and form a unique helical sheet that wraps around the M-domain. TM1 connects the Q- and M-domains and is only present in TRAP transporters such as HiSiaQM, where the Q- and M-domains are fused. The M-domain consists of TM helices 2–11 and the two hairpins HP1 and HP2. Structurally, TM 2–4 and TM 7–9 form a cradle around a bundle of helices formed by TM 5, 6, 10, 11, and HP 1, 2. This domain has an internal 2-fold pseudosymmetry that relates TM helices 2–6 and HP1 to TM helices 7–11 and HP2. While the periplasmic side of the transporter has a relatively flat surface, there is a deep cavity on the cytosolic side and the elevator domain protrudes into the cytoplasm, suggesting that the transporter was captured in its inward conformation ($C_i$-state) (Fig. 1).

Consistent with the observation that the Q- and M-domains form a tight complex even in the absence of the connection helix 1a, as seen in VcSiaQM[7], the two domains interact via a multitude of hydrophobic side chains such as W10, F16, F20, F48, F70, F98, F101, F125 on the Q-side and W212, F215, L237, V238, Y239, F244, F251, F455, M462 on the M-side of the interface. One of the few polar interactions in this interface is an ionic interaction between K45 and D242.

**TRAP transporters are monomeric elevators**. The closest structurally described homolog of the QM-domains is the dimeric VcINDY membrane transporter (5UL9)[39], a dicarboxylate transporter from *V. cholerae*, which operates via an elevator mechanism[30–32], now known to be employed by a diverse set of transporters[29,40–42]. The superposition of a VcINDY dimer onto our HiSiaQM structure illustrates that the M-domain fits to one VcINDY subunit and that the Q-domain of HiSiaQM structurally mimics the oligomerization domain of the second polypeptide

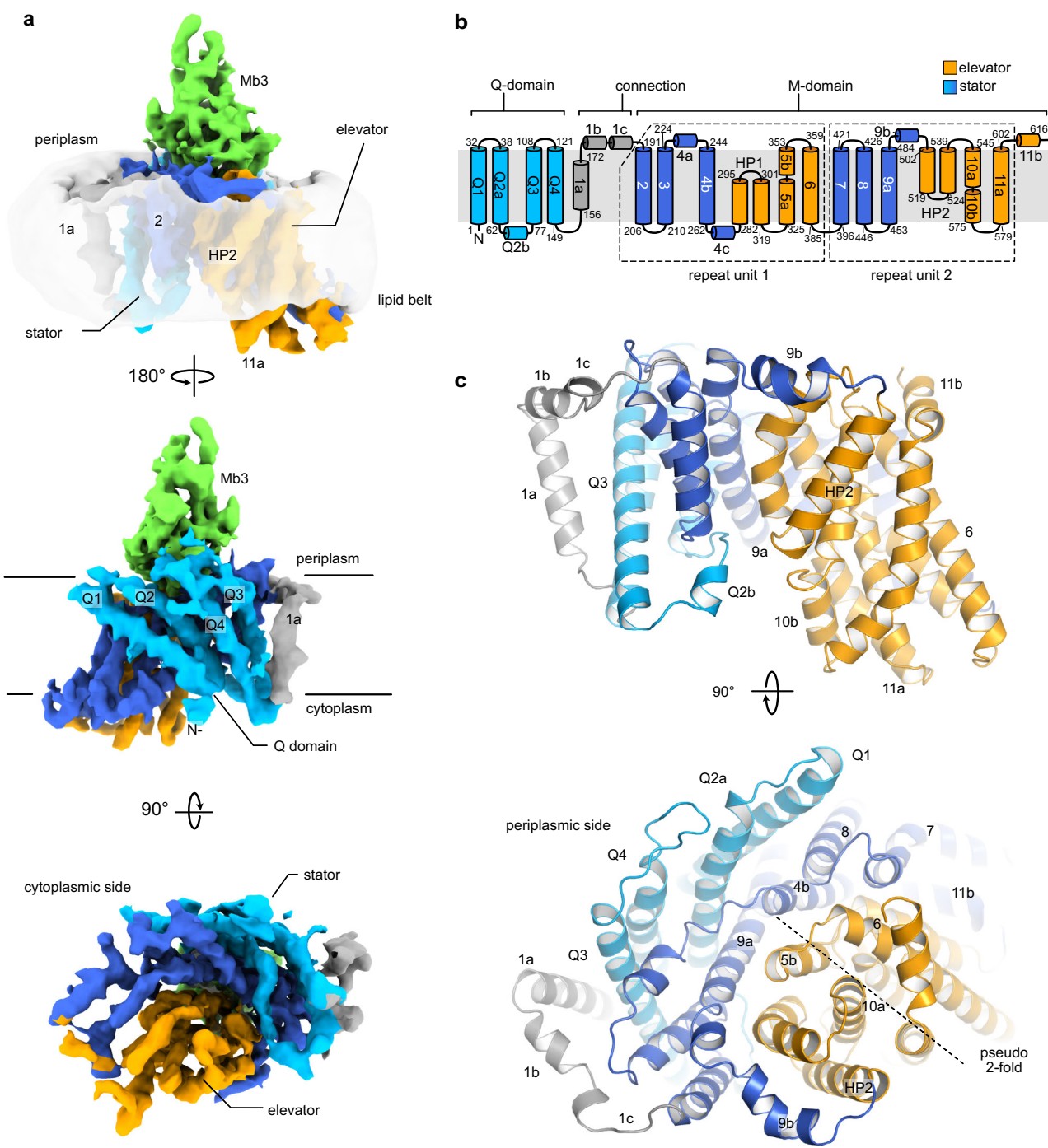

**Fig. 1 Cryo-EM structure of the TRAP transporter HiSiaQM. a** 3D reconstruction of HiSiaQM in complex with Mb3 in lipid nanodiscs (contour level 0.384). A map of the nanodisc was overlayed in transparent grey to show the relative position of the lipid bilayer. Selected secondary structure elements are indicated. The individual domains and their color code are identified in panel (**b**). **b** Topology diagram of HiSiaQM. **c** Cartoon representation of HiSiaQM with the same color scheme as in (**a**) and (**b**). Secondary structure elements are indicated.

chain of the VcINDY dimer (Fig. 2a). The four helices of the Q-domain are, however, significantly longer than their structural counterparts in VcINDY (Fig. 2b). The superposition also suggests that the substrate and co-transported sodium ions[8] are likely coordinated in the area surrounding the tips of HP1 and HP2 (Fig. 2c). Note that our in-lipid structure and its interpretation[43] has since been confirmed by a cryo-EM structure of the sialic acid TRAP transporter PpSiaQM from *Photobacterium profundum* that was determined in an amphiphile environment[44].

The internal pseudo-symmetry of the elevator fold allows the construction of the outward open (C$_o$) state of the transporter by "repeat-swap modeling"[31,45–47]. This procedure has for instance been used to build an inward-facing model of the GltPh transporter[46] and of the outward open state of the VcINDY transporter[31,48]. Using this model as a basis, we constructed the outward open state of the HiSiaQM transporter by structural alignment of the elevator and stator domains (Fig. 2d). Notably, the procedure did not produce any clashes in our model and

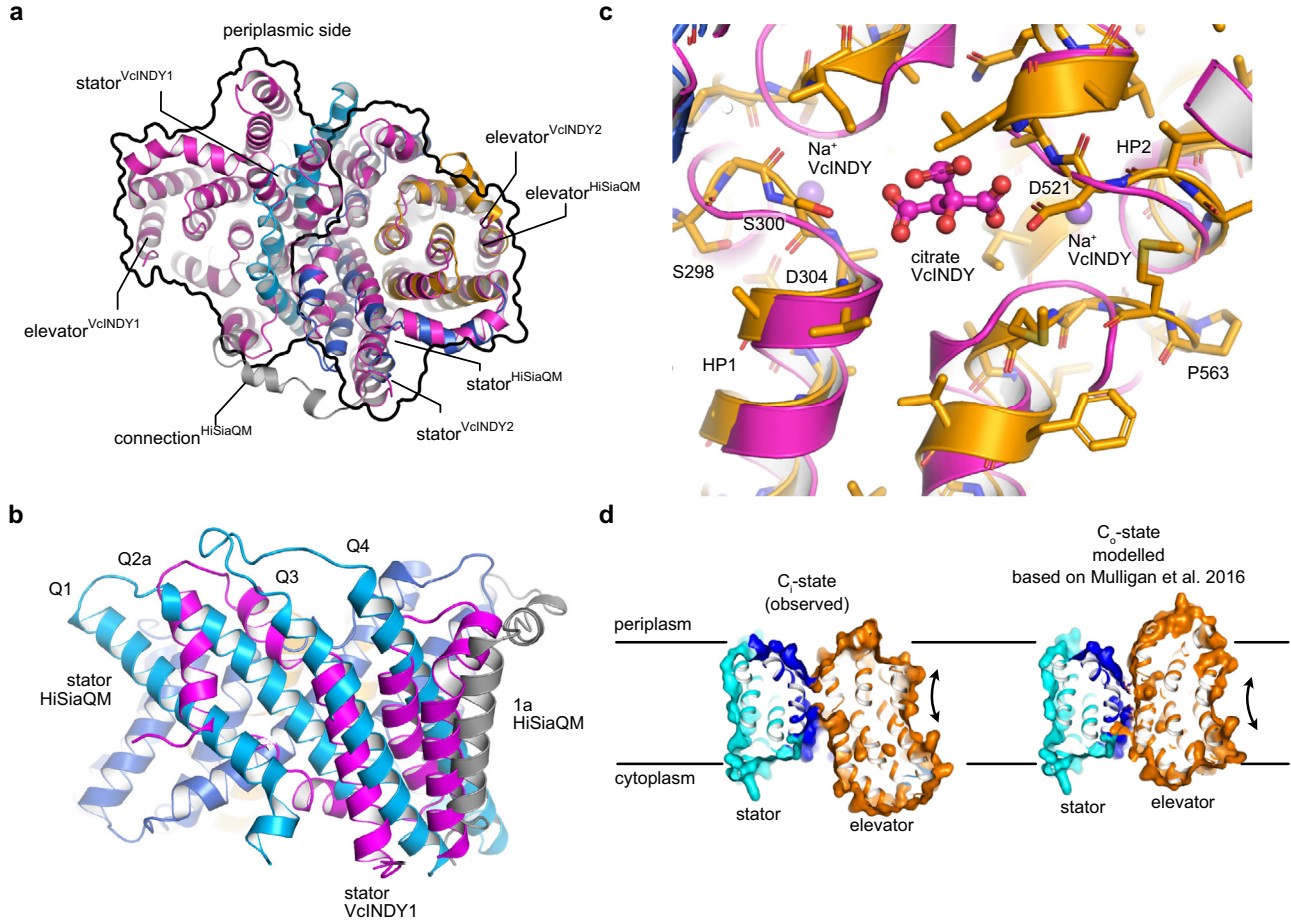

**Fig. 2 Comparison of the TRAP transporter HiSiaQM with elevator-type transporters. a** Superposition of HiSiaQM (colored coded as in Fig. 1) onto one unit of the dimeric VcINDY transporter in its $C_i$ state (magenta, 5UL9[39]). The outline of the two VcINDY monomers is drawn as a black line. Note that the Q-domain of HiSiaQM clearly crosses the "border" between the two VcINDY monomers. **b** Close-up view of the Q1–Q4 helices from (**a**). The magenta helices are part of the VcINDY stator domain. **c** The substrate-binding site of VcINDY aligned to HiSiaQM, showing the two $Na^+$ ions and the citrate molecule in the VcINDY structure. **d** Side view of HiSiaQM in the $C_i$ state from Fig. 1c (left) and a model of the outward open $C_o$ state (right) that was constructed based on the $C_o$ state of VcINDY[31,45–47].

agrees well with the predicted tripartite complex described below. The loops connecting elevator and stator domains were adjusted manually using the geometry regularization feature in Coot[49] (Supplementary Data 1).

A structural alignment of our in-lipid cryo-EM structure with the AlphaFold model results in an r.m.s.d. of 2.1 Å between 605 C-alpha atoms. Restricting the alignment to the Q-domain results in a smaller r.m.s.d. of only 1.2 Å between 136 C-alpha atoms. The differences between the two structures can be best described as a concerted movement of the elevator domain with respect to the stator domain with the elevator assuming a slightly more "downwards" conformation in our experimental structure vs. the computational prediction (Supplementary Fig. 8).

**High-affinity VHHs inhibit sialic acid transport.** An alpaca was immunized with DDM solubilized HiSiaQM, and nine distinct VHHs ($VHH_{QM}1$–9) were isolated. Using surface plasmon resonance (SPR) single-cycle kinetic experiments, high affinities in the nano- to picomolar range were confirmed for seven VHHs (Fig. 3a, Supplementary Fig. 9, Supplementary Table 2). Additionally, an epitope binning experiment revealed that the VHHs bind to at least two different regions of the transporter (Fig. 3a and Supplementary Fig. 9). The affinities depended on the immobilization site of the TRAP transporter (E235C-biotin or

K273C-biotin), which can be explained by our structure, since the biotinylated residues are located on different sides of the transporter (Fig. 3a).

Depending on their binding epitope, VHHs can influence the function of their target proteins, and examples of transport-inhibiting VHHs are known, also for elevator-type transporters[50–53]. To test this possibility for our HiSiaQM VHHs, we used a modified in vivo transport assay for TRAP transporters based on E. coli strain SEVY3[8,54]. In this strain, the native sialic acid transporter NanT was functionally replaced by HiSiaPQM, enabling it to grow in M9 minimal medium with sialic acid as the sole carbon source. We transformed different clones of the strain with our nine VHHs, either with or without a periplasmic export signal. In addition to the HiSiaQM VHHs, we included a camelid VHH for HiSiaP ($VHH_P1$) with an affinity of 0.89 µM to the SBP (Supplementary Fig. 10), since inhibition of SBPs by VHHs was described before[52]. Growth curves of the different strains were recorded to investigate the effect of the VHHs on cell growth (Fig. 3b). No significant inhibition of cell growth was observed when the HiSiaQM-specific VHHs were expressed without the signal sequence and thus remained in the cytosol. In contrast, when HiSiaQM-specific VHHs were exported to the periplasm, bacterial growth was strongly inhibited by $VHH_{QM}3$, 4, 6, 7, and 9. Cultures with $VHH_{QM}1$, 8, and $VHH_P1$ did show normal cell growth, irrespective of their cellular localization. As a control, we

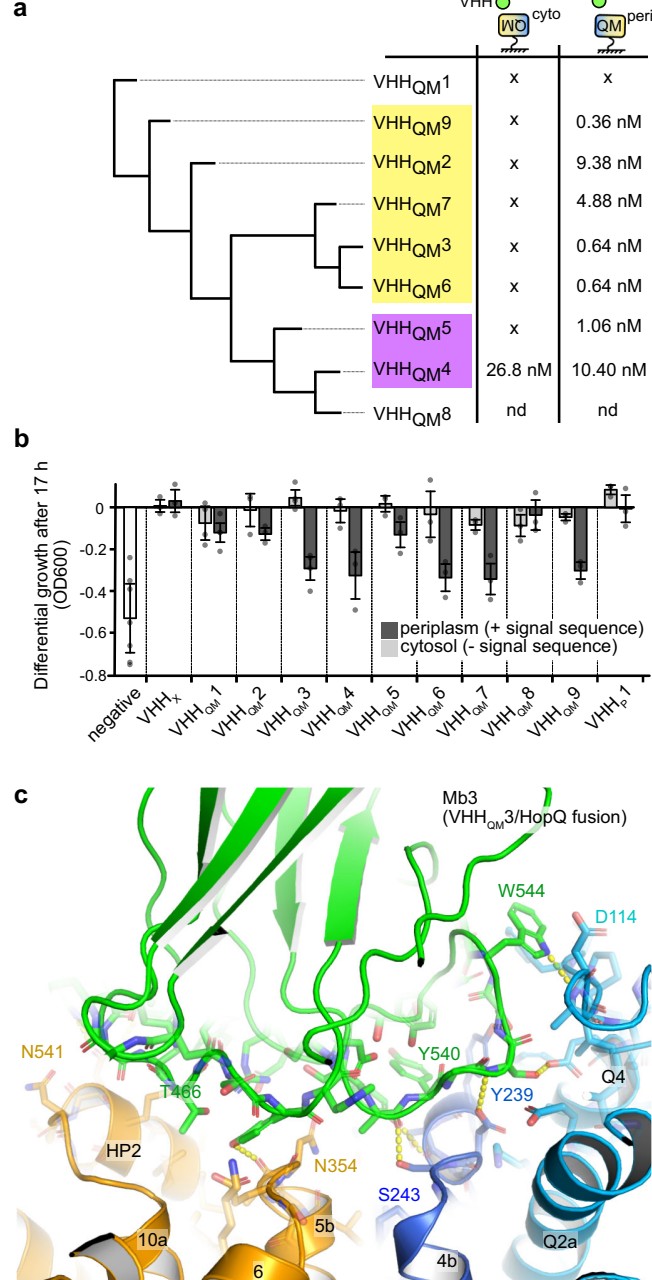

**Fig. 3 Characterization of TRAP transporter-specific VHHs and inhibition of transport in vivo. a** A hierarchical clustering tree of nine HiSiaQM-specific VHHs, based on PBLAST e-values as the distance matrix for tree building. The binding affinities of the VHHs, determined from SPR experiments are given (x: no binding detected; nd: not determined, since no clear binding detected in size-exclusion chromatography). VHHs that bind to HiSiaQM mutually exclusively are grouped by yellow and violet boxes. The underlying data are described in detail in Supplementary Fig. 9. HiSiaQM was immobilized on the SPR chip in two different orientations as indicated. **b** Growth defect of cultures expressing different VHHs specific for either the QM-domains (VHH$_{QM}$) or the P-domain (VHH$_P$) in a sialic acid uptake assay. The VHHs were either localized in the cytosol or exported to the periplasm via a signal sequence, as indicated in the figure. An empty plasmid was used as a negative control. A VHH specific for a completely unrelated human protein (VHH$_X$) was used as an additional control. The growth of each culture was measured after 17 h. Data are presented as mean values ± SD of $n \geq 3$ independent experiments. **c** Interface of HiSiaQM and Mb3, which derives from VHH$_{QM}$3. The color scheme is the same as in Fig. 1. Selected polar interactions and residues are highlighted. Source data are provided as a Source Data file.

is no clear correlation between the individual expression level and inhibitory effects, we cannot exclude that the level of inhibition of the individual VHHs is to an extent biased by their expression level (Supplementary Fig. 9d–g).

Since Mb3 was constructed from VHH$_{QM}$3, this information experimentally identifies the TM orientation of our cryo-EM structure. The molecular interaction between VHH$_{QM}$3 and the QM-domains is resolved in our structure (in the form of Mb3) and explains the inhibitory effect of this particular VHH (Fig. 3c), because it binds to both the elevator and stator domains of the transporter by forming multiple hydrogen bonds and a prominent hydrophobic interaction between W512 of Mb3 (residue 106 in VHH$_{QM}$3) and P112$_Q$, F111$_Q$ on the periplasmic loop Q3–Q4 of HiSiaQM (Supplementary Fig. 5f). These observations are further supported by the constructed tripartite model and the single-molecule data presented below.

**Constructing a model of the tripartite transport complex.** Encouraged by the close resemblance of the HiSiaQM AlphaFold model to our experimental structure (Supplementary Fig. 8), we employed the algorithm to predict the tripartite complex between HiSiaP and HiSiaQM by fusing the two proteins into a single chain[59]. The biological rationale behind this approach was the observation that in rare cases, natural M-P fusions do occur in nature, for example, a TRAP transporter from *Acidaminococcus intestini* (Uniprot ID: G4Q5D7) (Supplementary Fig. 11). As fusion peptides for HiSiaPQM, we used both the native variant from the *A. intestini* TRAP transporter and spacer residues from AlphaFold (U) (Supplementary Fig. 11). Both modelled HiSiaPQM variants, as well as the AlphaFold model of the natural M-P fusion transporter from *A. intestini*, were very similar to each other and to the previously mentioned model by Ovchinnikov et al. [38]. In all tripartite models, the SBP with its closed substrate-binding site (for HiSiaP most similar to 3B50[22]) is positioned in the same orientation on top of the periplasmic side of the membrane domains, which are in the inward-open conformation as found in our experimental structure (Fig. 4a and Supplementary Fig. 11, Supplementary Data 2). An analysis with the PISA server[60] revealed that a combined surface area of ~1980 Å$^2$ is buried by complex formation. Figure 4b shows the conservation of this P–QM interface across a large number of different TRAP transporters. In the tripartite HiSiaPQM complex, the N-terminal lobe of HiSiaP is bound to the stator domain of

repeated the same experiments with VHH$_X$, which is specific for an unrelated human protein and indeed showed no effect on cell growth. The results are in line with the SPR data, where no binding was detected to VHH$_{QM}$1 and 8 (Fig. 3a). Possible reasons for the lack of any effect of the VHH$_P$1 after periplasmic export are that the VHH binds to a site of HiSiaP, which is not involved in complex formation with HiSiaQM or in substrate binding, or that the high copy number of the SBP can simply not be saturated by the VHH[55,56]. We conclude that VHH$_{QM}$2, 3, 4, 5, 6, 7, and 9 bind to the periplasmic side of HiSiaQM.

To test for the possibility that the low inhibitory effect of VHH$_{QM}$2, 5, and 8 was merely due to the much lower expression level of these VHH$_{QM}$s relative to the VHH$_{QM}$7 construct (strongest inhibiting effect), we verified their expression by Western blotting (with and without the pelB export signal, Supplementary Fig. 9d–g). All VHHs were clearly expressed, but slight differences in the expression levels were indeed observed, which is a common finding for VHH expression[57,58]. While there

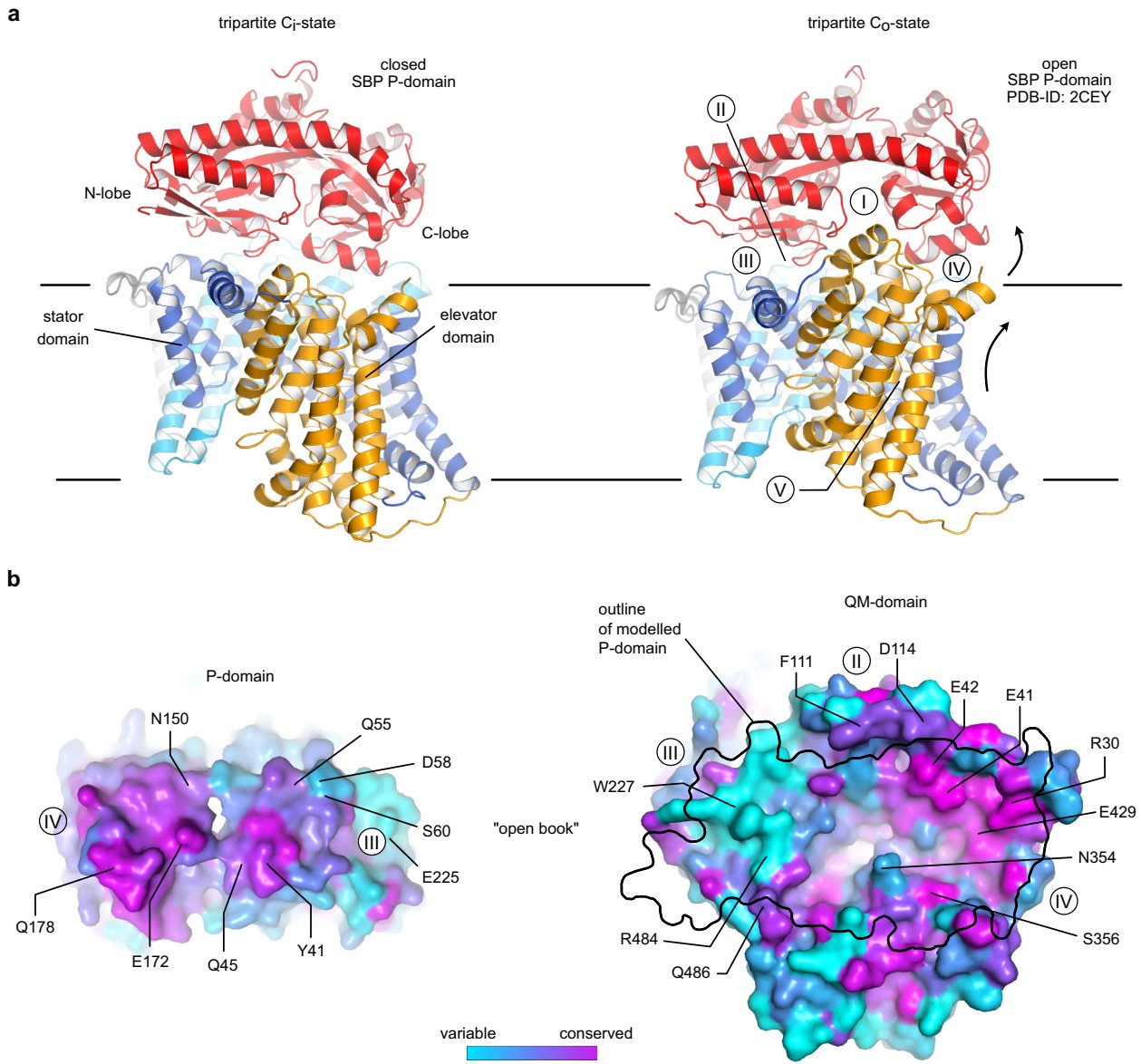

**Fig. 4 Constructing the tripartite PQM complex. a** Left: AlphaFold model of the tripartite complex between HiSiaQM and HiSiaP in the $C_i$ state. Right: The outward open model of HiSiaQM ($C_o$ state, Fig. 2d). Here, the P-domain was replaced by the open (substrate-free) structure of HiSiaP (2CEY) by aligning it onto the N-terminal lobe of the P-domain in the $C_i$ state (left). **b** "Open book" view of the interface regions between QM and the N- (III) and C-lobe (IV) of the P-domain. The color code corresponds to the conservation of residues, calculated by Consurf[85]. The highlighted residues are in regions I–V that are likely important for complex formation and were analyzed in the growth assay in Fig. 5.

HiSiaQM, and the C-terminal lobe is bound to the elevator domain. This arrangement puts the substrate-binding cleft of HiSiaP directly on top of the presumed substrate-binding site of the transporter, as identified by superpositions with substrate-bound VcINDY (Supplementary Fig. 12). This suggests that the N-terminal lobe of the SBP stays fixed on the stator, while its C-terminal lobe can move in concert with the up-and-down movement of the elevator during a transport cycle. Indeed, superposing the N-terminal lobe of the open-state crystal structure of HiSiaP (2CEY) onto the stator domain in precisely the same manner puts its C-terminal lobe into a position that closely matches the elevator in its above-described outward open conformation (Fig. 4a). The relative orientation of the P-domain to the QM-domains in our models is compatible with TRAP SBPs that are known to form stable homodimers, for example, TakP from *Rhodobacter sphaeroides* (Supplementary Fig. 13)[61,62]. The

dimeric P-domains superpose such that the second monomers point upwards, not interfering with the interface of the tripartite P-QM complex.

**Validation of the tripartite model with an in vivo sialic acid uptake assay.** To validate the described models, we selected 31 residues in regions that we thought were important for the integrity and function of the TRAP transporter, such as the substrate-binding site of the P-domain (region I), the extended periplasmic loops of the Q-domain (region II), the P–QM interface (regions III and IV), or the assumed sialic acid- and $Na^+$ binding sites at HP1 and HP2 (V) of the QM-domains (Fig. 5a) (detailed views of the mutation sites are shown in Supplementary Fig. 14 and the sequence conservation of TRAP transporters is shown in Supplementary Figs. 15 and 16). Highly conserved sites

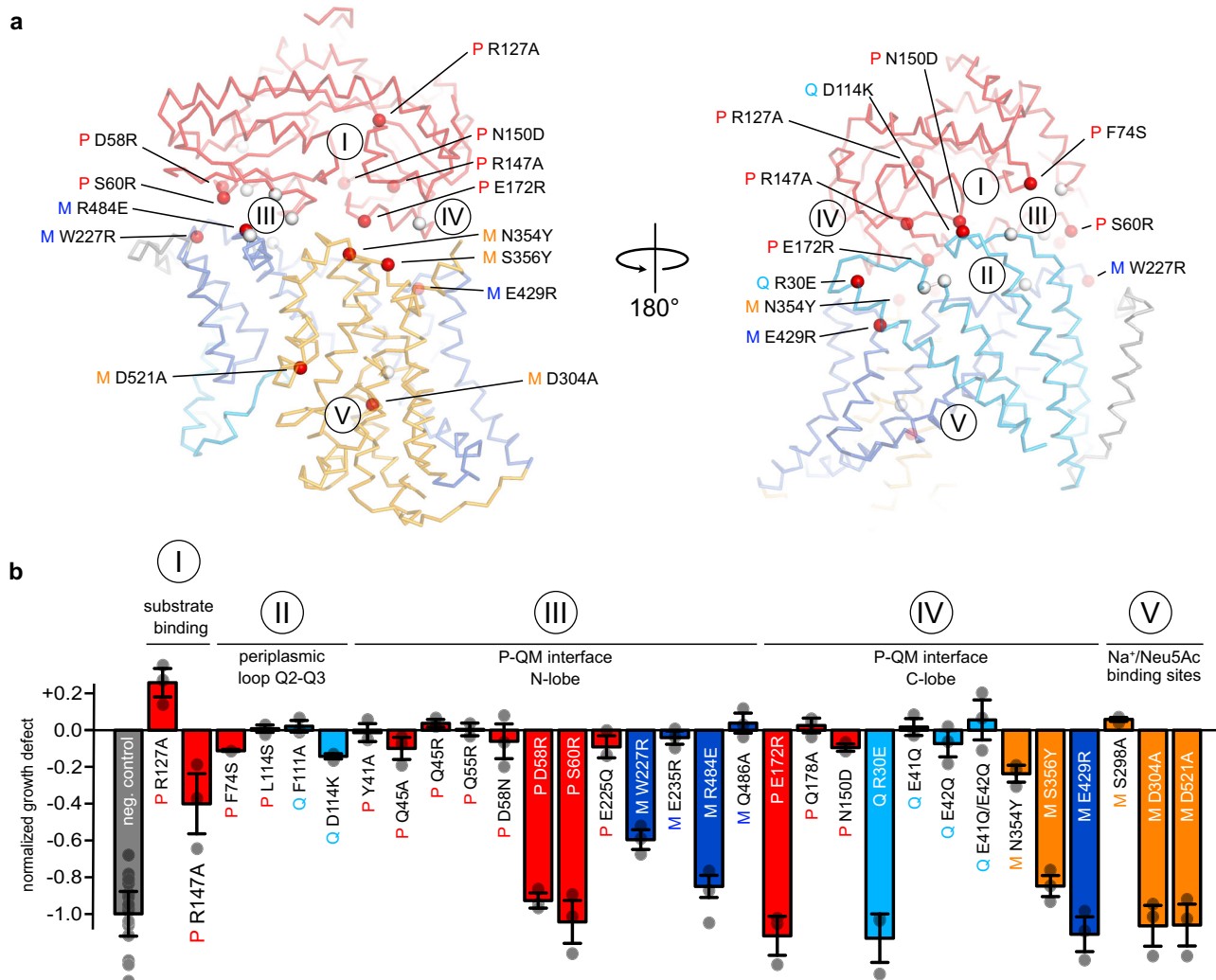

**Fig. 5 Validation of the tripartite P-QM complex. a** Structural context of the mutations in the AlphaFold model of the tripartite complex. The P-domain is colored red and the QM-domains are color-coded as in Fig. 1. Red spheres indicate mutants that showed a significant effect in the growth assay and are labelled. White spheres represent the remaining mutants. **b** Growth defect after 17 h (compared to a wildtype culture and normalized to the negative control) of TRAP transporter mutants in a sialic acid uptake assay. The positions of the mutants with respect to the tripartite complex are indicated by labels and the color code. Data are presented as mean values ± SD of $n \geq 3$ independent experiments. Source data are provided as a Source Data file.

in the periplasmic loops of the Q-domain (region II) were selected to test for a potential "scoop-loop" mechanism, as found in SBP-dependent ABC transporters[63]. The effects of all mutants were analyzed in our SEVY3-based complementation assay.

We identified 10 mutants that severely or completely abolished growth and are structurally spread over all three transporter domains (Fig. 5a, b). Mutations of highly conserved residues $R127A_P$ and $R147A_P$ in the P-domain (region I, Fig. 5), which are known to be important for substrate binding and were previously described as a substrate-filter[25], showed comparably small effects in the growth assay. This observation is in accordance with previous data and crystal structures of substrate-bound $R147_P$ variants, where the arginine–carboxylate interaction is mimicked by water molecules[25]. It seems that at high substrate concentrations (3.2 mM in our assay) this can be tolerated by the system. Minor effects were also observed for mutants of the periplasmic loops or the adjacent region of the P-domain (region II, Fig. 5).

Twelve mutants are located in region III (Fig. 5), which connects the N-terminal lobe of HiSiaP and the stator of HiSiaQM. Here, the most substantial effect on growth was observed with mutants $D58R_P$ and $S60R_P$. Due to their larger size

and reversed charge, the introduced side chains would likely interfere with a positively charged surface patch of QM, explaining their effect. This is corroborated by our finding that the conservative $D58N_P$ mutant did not show any effect on growth. Note that all P-domain mutants that failed to form a functional transporter in vivo, showed wildtype-like binding behavior towards sialic acid in ITC experiments, strongly indicating that the mutated residues are indeed essential for the tripartite complex formation and do not otherwise impair the function of the P-domain (Supplementary Fig. 17). The $R484E_M$ position is also part of region III. While the glutamate sidechain can be accommodated structurally, the charge reversal appears to impact complex formation, albeit the effect on growth was not as strong as for the former two mutants that come from the "P-side" of the interface. Position $W227R_M$ was selected to modify the interface indirectly by pushing TM helices 3 and 4a into a different position by introducing a large charged residue into the hydrophobic pocket occupied by $W227_M$, $F101_Q$, $V230_M$, and $F177_M$. Indeed, this also led to a substantial reduction in growth.

Ten of the tested mutants are located in region IV, formed by the C-terminal lobe of HiSiaP and the elevator domain of

HiSiaQM. In this region, the strongest effects were seen with mutants $E172R_P$, $R30E_Q$, $S356Y_M$, and $E429R_M$. The variants are spread over a slightly larger area than the interface III mutants. $E172R_P$ and $S356Y_M$ would again lead to severe steric clashes. The effect of the $R30E_Q$ mutant of the strongly conserved $R30_Q$ cannot be explained by steric hindrance in the $C_i$-state of the transporter. However, we noticed that this residue is in the salt-bridge distance with the other knock-out mutant $E172R_P$ in the $C_o$-state transporter model (Fig. 4). In addition, the highly conserved $E429R_M$ is located directly next to $R30_Q$, and the strong growth defect of this mutant supports the high importance of the region for the interaction of the domains. A weaker but significant growth reduction was observed for $N354Y_M$. In our model, the asparagine forms hydrogen bonds with $E172_P$ in the $C_i$-state of the transporter.

For the potential $Na^+$-binding site mutants (region V) two of three mutants ($D304A_M$ and $D521A_M$) independently led to complete loss of transporter function, providing a structural explanation for the earlier finding that a sodium ion gradient is strictly needed for transport by HiSiaPQM[8]. Since the sodium substrate-binding sites are spatially linked, the effect might also stem from an altered affinity of the transporter to sialic acid.

All mutants that completely inhibited growth in our uptake assay were expressed and purified. As mentioned above, the P-domain mutants had wild-type-like affinity for sialic acid and all QM-mutants behaved overall similar to the wildtype protein in size-exclusion chromatography experiments (Supplementary Fig. 18).

**Tracking the formation of the tripartite transport complex in vitro.** We employed single-molecule TIRF microscopy to visualize the tripartite transport complex in solid-supported DOPC bilayers (SSBs[64]), allowing us to directly study the impact of sialic acid and the above-described P–QM interface mutants on the complex formation (Fig. 6a).

The native QM-domains were reconstituted into SSBs and an AF-555-labelled $VHH_{QM}3$ (Fig. 6a) was used to identify and localize the reconstituted transporter in the lipid bilayer (Fig. 6b, Supplementary Movie 1). The VHH was also used to optimize the experimental conditions, such as the number of QM-domains in the field of view of the microscope. No unspecific binding of $VHH_{QM}3$ to the bilayer was detected in control experiments with empty SSBs (Fig. 6c).

Next, we incubated QM-SSBs with AF-647-labelled P-domains (Fig. 6a, final concentration of ~100 pM). Strikingly, we could clearly observe single P-domains appearing out of the bulk solution and localizing to defined points on the membrane (Fig. 6d, Supplementary Movie 2). This direct experimental observation of the formation of the full tripartite complex confirms a key step in the transport cycle.

To investigate the role of sialic acid in the P–QM interaction, we omitted the compound from the SSB TIRF experiment. To our surprise, there was only a ~50% reduction of P–QM binding events in the absence of sialic acid (Fig. 6e).

Both $VHH_{QM}3$ and the P-domain bind to the same position on the periplasmic surface of the transporter (Fig. 6a). Hence, the VHH should block the P–QM interaction in the SSB experiments, as indicated by our in vivo experiments (Fig. 3). To test this in vitro, we preincubated the QM-SSB with unlabeled $VHH_{QM}3$ and then added the AF-647-labelled P-domain. In this experiment, only a very small number of binding events could be tracked (Fig. 6f). The number of observed interactions was in fact comparable to a control experiment with empty SSBs, which also proved that the P-domain does not bind to the bilayer in an unspecific manner (Fig. 6g).

Finally, we investigated 9 different mutants which had a clear loss of function phenotype in the sialic acid uptake assay (Fig. 5). For the three P-domain mutants $D58R_P$, $S60R_P$, and $E172R_P$ the number of binding events was drastically decreased to levels comparable to the experimental background for $D58R_P$ and $E172R_P$ (Fig. 6h, j) and a slightly higher number of events for $S60R_P$ (Fig. 6i). As mentioned above, all three mutants had a wild-type-like affinity towards sialic acid (~50 nM) (Supplementary Fig. 17), which strengthen the hypothesis that the introduction of large, charged arginines disrupt the tripartite complex formation. Similar results were observed for the four QM-domain mutants $R30E_M$, $S356Y_M$, $E429R_M$, and $R484E_M$, which are all located on the periplasmic site in the P-domain interacting region, and clearly showed no significant binding events compared to the experimental background (Fig. 6k–n, Supplementary Movies 2 and 3). Interestingly, the two mutants from the sodium binding site, which had no transport activity in the sialic acid uptake assay, showed a wild-type-like interaction with the P-domain (Fig. 6o, p).

The TIRF data above is supported by surface plasmon resonance (SPR) experiments where HiSiaQM (in DDM micelles) was immobilized on an SPR chip as described above in the VHH-binding studies. Figure 6r shows that the closed P-domain interacted with the immobilized transporter (5 mM sialic acid was added to the running buffer). The interaction was clearly weaker in the absence of sialic acid and could be blocked by saturating the immobilized QM-domain with $VHH_{QM}3$ (Fig. 6r, s). Interestingly, the resulting sensorgrams and especially the fast dissociation of the analyte could not be satisfyingly fit with a 1:1 binding model (Fig. 6r). Hence, the true $K_D$ is very likely higher (i. e. weaker) than the average $K_D$ of ~1 µM that was determined from a set of four independent SPR experiments. The deviation from the 1:1 model indicates a complex binding behavior, which is maybe not surprising considering that at least two separate binding interfaces (stator/N-lobe and elevator/C-lobe) are involved, combined with the conformational flexibility of the participating molecules. Similar observations of complex binding events in transport mechanisms have been made with the BtuCD-BtuF ABC transporter[65] or the GltPh elevator-type transporter[66]. Clearly, more detailed experiments are needed to fully understand the binding kinetics of the tripartite complex. However, taken together, the TIRF and SPR data support our observation from the complementation assay and our tripartite HiSiaPQM complex model.

## Discussion

Our study reveals the unique structural architecture of TRAP transporters and provides important insights into their function. Based on the above-described structures, and in accordance with our biochemical-, microscopy- and in vivo data, as well as with previously published results, we suggest that TRAP transporters employ an elevator mechanism for substrate translocation. Strikingly, TRAP transporters are monomeric elevator-type transporters, expanding the landscape of transporter architecture known in biological systems. Our structural data indicate that the characteristic Q-domain of TRAP transporters, whose essential function has been a mystery for many years, is central to this architecture, by representing a structural mimic of the combined stator domains of multimeric elevators (Fig. 2). We hypothesize that a minimal size of the stator is needed to anchor the domain in the membrane and to support the up-and-down movement of the elevator domain. The extended length of the Q-domain helices in HiSiaQM compared to their structural counterparts in VcINDY might further stabilize this asymmetric design. While the bile acid transporter ABST is also monomeric,

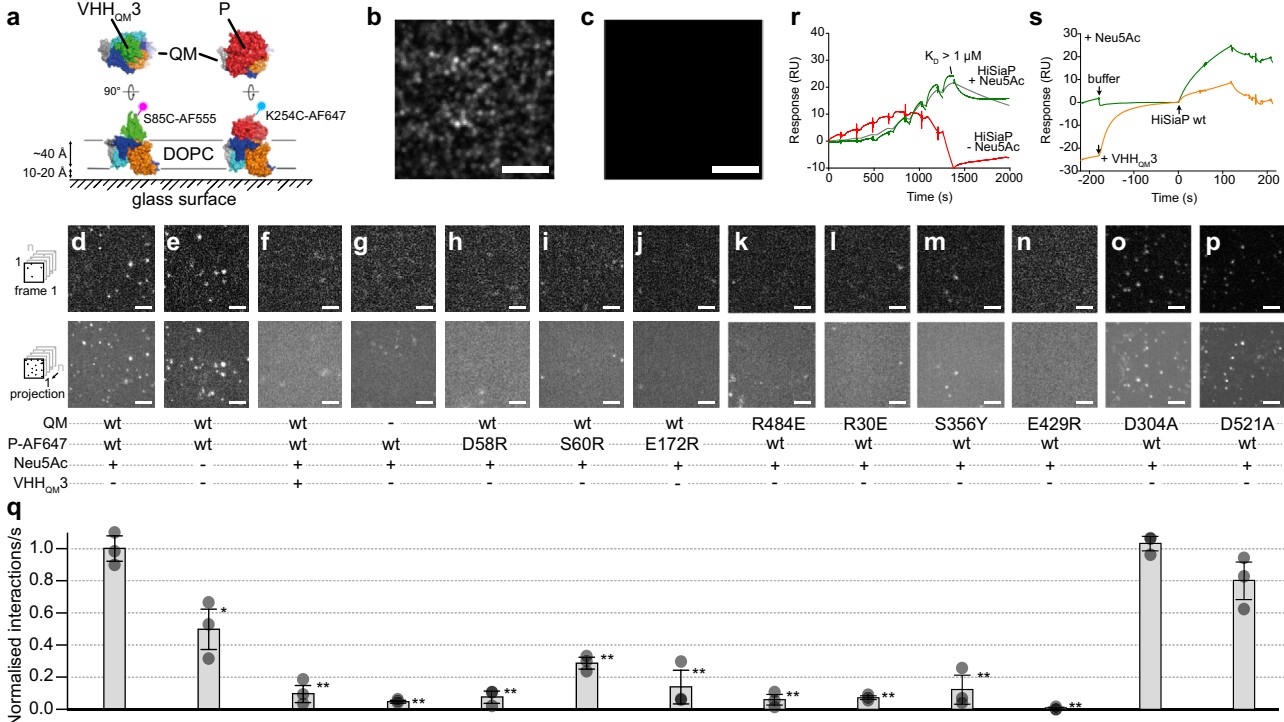

**Fig. 6 Single-molecule interaction studies of HiSiaQM and HiSiaP on solid-supported bilayers (SSBs). a** HiSiaQM variants were integrated into DOPC (1,2-Dioleoyl-sn glycero-3-phosphocholine) SSBs and their interaction with single AF-647-labelled P-domain variants or AF-555-labelled VHHQM3 was observed by TIRF microscopy. **b** SSB containing HiSiaQM visualised with AF-555 labelled VHHQM3. **c** As a control, VHHQM3-AF-555 was added to an SSB in the absence of HiSiaQM. No unspecific binding was observed. **d–p** Top row: first frame of an image sequence of a typical set of data. Bottom row: maximum intensity projections of the respective image sequence. The conditions are indicated below each panel. **q** Normalised interactions per second of P-domain variants with a SSB containing HiSiaQM variants. Unless otherwise stated, the P-domains were pre-incubated with 10 mM sialic acid (Neu5Ac). Statistical significance of the P-mutants and controls was assessed by applying a two-sided unpaired Student's $t$-test with a 95% confidence interval (*$p < 0.01$, **$p < 0.001$). The full dots represent the average results of $n = 3$ independently prepared samples, respectively. The bars represent the mean value of the three averages. The distribution of the full dots indicates for every condition the reproducibility of the results. For each of these three samples, a fresh bilayer was prepared on a new coverslip and $n = 30$ individual measurements were performed. The underlying data is compiled in Supplementary Table 3. The scale bars equal 3 μm. **r** Sensorgrams showing the interaction of the P-domain with immobilised QM-domains (HiSiaQM K273C-biotin in DDM) in the absence (red) or presence (green) of sialic acid (5 mM). The green curve was fit with a 1:1 binding model (grey). Four independent experiments were performed. **s** Competition experiment between the P-domain and VHHQM3 for the immobilized QM-domains. Green curve: a buffer injection followed by an injection of the P-domain. Orange curve: the chip surface was saturated with VHHQM3 before the P-domain was added. Single experiment, supported by multiple other experiments in this study (e.g. panel (**f**), Fig. 3b, the cryo-EM structure). Source data are provided as a Source Data file.

it uses a mixture of the moving barrier and elevator mechanisms[29,67] and a structurally different elevator mechanism has also been postulated for the CcdA transporter[68], the SiaQM structure uniquely represents a monomeric version of a classical elevator.

Interestingly, the likely substrate-binding site formed by the QM-domains does not contain any conserved positively charged residues, such as in the SBP, where Arg147 determines the specificity for its negatively charged monocarboxylate substrate[25]. In contrast, the only other structurally characterized sialic acid transporter SiaT from *Proteus mirabilis* (5NVA), a structurally unrelated sodium-solute symporter (SSS) transporter, does have a conserved arginine in its translocation channel, and this residue interacts with the substrate during transport[69]. Since SiaT does not use an SBP, this supports a model where the substrate selectivity of TRAP transporters is "outsourced" from the membrane domains to the SBP, explaining why the latter is strictly needed for TRAP transporter-mediated transport[1,7,8].

We used AlphaFold[37] to generate a structural model of the tripartite PQM transporter complex. The resulting model elegantly rationalizes the results of our in vivo mutagenesis study and is in very good agreement with our single-molecule tracking

microscopy data (Figs. 5 and 6). It is further supported by our finding that VHHs, which bind to the periplasmic side of the QM-domains efficiently block transport activity (Fig. 3). The observation that the crystal structures of the closed- and open states of HiSiaP comfortably fit our inward open structure of HiSiaQM and a model of the outward open transporter, respectively, indicates that not only the interfaces of these two proteins but also their conformational flexibility have been matched by evolution. Clearly, this has to be proved by experimental structures of the P–QM complex in the future.

The information presented above allows us to amend the working hypothesis for TRAP transporter-mediated transport that was last updated by Mulligan et al. in 2011[4]. As known from several previous studies, the SBP binds tightly ($K_D = 20$–300 nM depending on the SBP) to its substrate and switches from an opened- to a closed conformation (Fig. 7, step 1)[18,20,27]. As the P-domain only closes when a substrate is bound, empty transport cycles are prevented[21,27]. Both, the TIRF and SPR data show that the presence or absence of sialic acid and, accordingly the conformational state of the P-domain, strongly impacts the formation of the tripartite complex (Fig. 6, Supplementary Fig. 19). The TIRF experiments in lipid bilayers demonstrate that the opened

**Fig. 7 Proposed mechanism of TRAP transporters.** The schematic shows the components of the transport reaction in the different steps (numbers in circles) of the proposed transport mechanism, which are explained in the main text. The conformational state of the QM domains is annotated: $C_i$—inward open; $C_o$—outward open; $C_o$-S—outward open, substrate bound; $C_i$—S inward open, substrate bound. Neu5Ac—sialic acid (*N*-acetylneuraminic acid).

P-domain (i.e., in the absence of sialic acid) does interact with the transporter, albeit to a significantly reduced extent compared to the substrate-bound P-domain. Considering the architecture of the tripartite AlphaFold model (Fig. 4), this result may be explained by weak interactions of either the N- or C-lobe of the open P-domain with either the stator or elevator domain of the transporter. Since the C-lobe has much higher sequence conservation (Fig. 4b), one might speculate that this part of the open P-domain interacts with the transporter. Our observation explains a finding by Mulligan et al. [8] that the transporter can be forced to "run backwards" with a steep but "inverted" and thus non-physiological sialic acid gradient. Clearly, this requires a functional interaction of the apo P-domain with the transporter, consistent with our data.

In the next step, the substrate-bound SBP binds to the transporter in its inward open resting state, the structure of which has been determined in this work (Fig. 7, step 2). Our microscopy and SPR data are in agreement that the sodium gradient, which is necessary for transport[8], is not needed for this step of the transport cycle since no gradient was present in either the SSB- or SPR experiments (Fig. 6, Supplementary Fig. 19). Binding of the SBP must then trigger the upward movement of the elevator so that the SBP is opened and its ligand released, as suggested for the TeaABC TRAP transporter[70] (Fig. 7, step 3). As the transport process is coupled to the translocation of $Na^+$ ions, this upwards movement is presumably accompanied by the binding of two such ions and the relaying of the substrate from the P-domain to the QM-domains[8] (Fig. 7, steps 3, 4). Dissociation of the open SBP presumably allows the elevator domain to "fall" back into the inward open state (Fig. 7, step 5). The substrate and the $Na^+$ ions then diffuse into the cytosol, resetting the transport cycle (Fig. 7, step 6).

Since TRAP transporters are absent in humans and play an essential role in the virulence of important pathogens, such as *H. influenzae* or *V. cholerae*, they are attractive targets for the development of new antibiotics. Our finding that TRAP transporters can be inhibited with several different VHHs opens a road in this direction and can be used in following studies, for example in the development of small compounds that mimic the CDRs of the VHHs and thus lead to a similar effect on transport, or by using them as tools in high throughput displacement assays.

## Methods

**Cloning, expression, and purification of HiSiaQM.** The *hisiaqm* gene (HI0147, Uniprot: P44543) was previously cloned into a pBAD vector with an N-terminal His$_{10}$-tag and a TEV-cleavage site[8]. HiSiaQM mutants were produced with site-directed mutagenesis[71] (Supplementary Table 4). For expression, the plasmid was transformed into MC1061 *E. coli* cells and a preculture of 100 mL LB-medium (100 µg/ml ampicillin) was grown overnight at 37 °C and 140 rpm. The expression was performed by inoculation of 10 l of fresh LB-medium (100 µg/ml ampicillin) with the preculture and grown for 2–3 h at 37 °C and 140 rpm until an OD$_{600}$ of 0.8. The protein expression was induced with 50 mg/l L(+)-arabinose and incubated for 2 h at 37 °C and 140 rpm. The cells were harvested by centrifugation at

4000 × g for 20 min and 10 °C. The cell pellets were stored at −80 °C until further use.

For purification, the cells were resuspended in 4 times excess 50 mM KH$_2$PO$_4$ (pH 7.8), 200 mM NaCl and 20% glycerol (buffer A). The cells were lysed by sonication (40% amplitude, 5 min, pulses 10 s on-5 s off) on ice and afterward centrifuged at 300,000 × g for 1 h at 10 °C. The supernatant was discarded and the pellet was resuspended in buffer A, supplemented with 1.5% (w/v) dodecyl-β-D-maltoside (DDM). After incubation overnight at 4 °C under gentle shaking, the solution was again ultra-centrifuged as before. The supernatant was mixed with Ni-NTA agarose beads, which were previously equilibrated with buffer A, and incubated for 2 h under gentle shaking at 4 °C. Afterwards, the suspension was loaded onto a benchtop column at 4 °C, the flowthrough was discarded and the beads were washed with 100 ml 50 mM KH$_2$PO$_4$ (pH 7.8), 200 mM NaCl and 0.035% DDM (buffer B) with 22 mM imidazole. The protein was eluted with 15 ml buffer B, supplemented with 250 mM imidazole, and concentrated to 0.5 ml with a Vivaspin (100 kDa MWCO). The protein was loaded onto an equilibrated Superdex 200 increase 10/300 column on an ÄKTA chromatography system and eluted in fractions from the column with buffer B. All purification steps and the fractions from the size-exclusion chromatography were checked with an SDS–PAGE and HiSiaQM containing fractions were combined, concentrated to around 15 mg/ml and flash-frozen for storage at −80 °C.

**Cloning, expression, and purification of HiSiaP.** The SBP was previously cloned into a pBADHisTEV vector which fuses the HiSiaP protein to an N-terminal His$_6$-tag and TEV-cleavage site. HiSiaP mutants were produced with site-directed mutagenesis[71] (Supplementary Table 4). The protein was expressed and purified as described in Peter et al. [21]: *E. coli* BL21 cells were used for expression and a 50 ml LB-medium culture was incubated overnight at 37 °C. On the next day, the culture was washed with M9 minimal medium to avoid contamination of HiSiaP with sialic acid during expression. For this, the culture was centrifuged and resuspended in 100 ml M9 minimal medium. After repeating the washing step once more, 20 ml of the resuspension was used to inoculate 1 l M9 minimal medium. The cells were incubated overnight for 14–16 h until OD$_{600}$ of 0.6–1.0, expression was induced with 500 mg/l L(+)-arabinose and the cells were harvested after 3 h at 37 °C. The pellet was stored at −80 °C or directly used for purification. For purification, the pellet was resuspended in 5 times excess of 50 mM Tris (pH 8), 50 mM NaCl and lysed with a sonicator on ice (40% amplitude, 5 min, pulses 10 s on-5 s off). After 20 min. centrifugation with 75,000 × g the supernatant was filtered and mixed with Ni-NTA agarose beads and equilibrated in a resuspension buffer. After incubation for 1 h at room temperature, the mixture was pipetted in a bench-top column and the flowthrough was discarded. After a wash step with 100 ml resuspension buffer, the protein was eluted with 15 ml elution buffer (50 mM Tris (pH 8), 50 mM NaCl and 500 mM imidazole). The imidazole was removed with several concentration and dilution steps in a Vivaspin (30 kDa MWCO) and the protein was incubated with TEV protease overnight at 4 °C (TEV protease:HiSiaP, 1:50 ratio). On the next day, the TEV protease and cleaved tag were removed with another Ni-NTA affinity chromatography and the flowthrough was collected, concentrated, and loaded on an equilibrated HiLoad Superdex 200 16/600 column (50 mM Tris (pH 8), 50 mM NaCl). Eluted fractions with protein were analyzed with SDS–PAGE and corresponding fractions were pooled, concentrated to around 30 mg/ml, flash-frozen and stored at −80 °C.

**Expression and purification of MSP1D1-H5.** The *msp1d1* gene was ordered from Addgene (#20061) in a pET28a vector which fused an N-terminal His$_6$-tag and TEV-cleavage site to the MSP1 protein. To decrease the size of the nanodiscs, helix 5 (residues L68-L87) was deleted via PCR[71] (Supplementary Table 4), yielding construct MSP1D1-H5[34]. The expression and purification were based on ref. [72]. In brief, the plasmid was transformed into *E. coli* BL21 cells and a preculture of 120 ml LB-medium (50 µg/ml kanamycin) was grown for 4–6 h at 37 °C and 140 rpm until an OD$_{600}$ of 0.6–0.8. The culture was stored at 4 °C overnight and used to inoculate 6 l TB-medium (50 µg/ml kanamycin) the next day. The culture was incubated at 37 °C and 140 rpm until an OD$_{600}$ of 2.5–3.0 and expression was induced with a final concentration of 1 mM IPTG in the culture. After expression of 1 h at 37 °C

followed by 3.5 h at 28 °C under shaking with 140 rpm, the cells were harvested by centrifugation at 4000 × g for 15 min. The cell pellet was stored at −80 °C until further use.

For purification of MSP1D1-H5, the cell pellet was resuspended in 2.5 times excess of lysis buffer (20 mM NaPi (pH 7.4), 1% Triton X and 0.001% nuclease). Cells were lysed by sonication for 5 min at 40% amplitude with pulses of 10 s on, 5 s off, under constant cooling on ice. After centrifugation of the lysate for 30 min at 30,000 × g and 10 °C, the supernatant was mixed with Ni-NTA agarose beads, previously equilibrated with basic buffer (40 mM Tris (pH 8.0), 300 mM NaCl) and incubated for 1 h at room temperature under gentle shaking. The suspension was transferred onto a bench-top column at room temperature and the flowthrough was discarded. The beads were washed in four steps of 12 ml with basic buffer, supplemented in the first step with 1% Triton X, in the second step with 50 mM sodium-cholate and no supplementation in the third step. Afterward, the protein was eluted from the column with 15 ml basic buffer with 300 mM imidazole and concentrated to a total volume of 5 ml. The eluate was dialyzed overnight at 4 °C against 100 times excess of buffer with 40 mM Tris (pH 8.0). For further purification, the protein solution was loaded onto a HiLoad Superdex 200 16/600 column on an ÄKTA chromatography system, previously equilibrated with 10 mM Tris (pH 7.4), 100 mM NaCl and 1 mM EDTA. The protein was eluted in fractions and checked via an SDS–PAGE for protein-containing fractions. Selected fractions were combined and incubated overnight at 4 °C with TEV protease in a 1:50 ratio. Afterward, the cleaved tag and TEV protease were removed via Ni-NTA affinity chromatography and cleaved MSP1D1-H5 was collected in the flowthrough. TEV-cleavage was checked via SDS–PAGE and the protein was concentrated and stored at −80 °C.

### Reconstitution of HiSiaQM in MSP1D1-H5 nanodiscs. The nanodisc reconstitution procedure for the TRAP transporter was based on ref. [72]. The purified MSP1D1-H5 and HiSiaQM were mixed in buffer C (50 mM KH$_2$PO$_4$ (pH 7.8), 200 mM NaCl), supplemented with 50 mM DMPC and 100 mM sodium cholate, to a ratio of 1:60:0.2 (MSP1D1-H5:DMPC:HiSiaQM). The solution was diluted with buffer C to a final concentration of 11 mM DMPC and 22 mM sodium-cholate and incubated for 2 h at 26 °C under gentle shaking. Afterward, the reconstitution mix was dialysed in a tube with 6–8 kDa MWCO against 500 ml buffer C at 26 °C for 16 h with 4 buffer exchanges.

### Generation, cloning, expression, and purification of VHHs. VHHs were generated by the Core Facility Nanobodies of the University of Bonn. All immunizations were authorized by the Landesuntersuchungsamt Rheinland-Pfalz (23 177-07/A 17-20-005 HP). One alpaca (Vicugna pacos) was immunized by six subcutaneous injections over 12 weeks with 200 µg DDM detergent-solubilized wild-type HiSiaQM mixed 1:1 (v:v) with GERBU-FAMA adjuvant. After the immunization peripheral blood mononuclear cells (PBMCs) were isolated from 100 ml blood. The RNA was extracted from PBMCs, and reversely transcribed to cDNA. VHH sequences were amplified by PCR using specific primers and cloned into a phagemid vector. E. coli TG1 cells were transformed with the VHH phagemid library and infected with helper phage VCSM13 to produce phage displaying VHHs as pIII fusion proteins. VHHs were enriched by phage panning on HiSiaQM E235C-biotin and K273C-biotin (see below) bound to streptavidin beads. E. coli ER2738 were infected with the enriched phages and after one round of panning, individual clones were grown in 96-well plates. VHHs in the supernatants were tested for specific binding by ELISA. Candidates were sequenced and grouped by sequence similarity. The HiSiaP-VHH was identified during phage display with a VHH library from a homologous VcSiaP immunization campaign with the same procedure as described above.

For protein expression, the genes were cloned into a pHEN6 vector which fused the protein to an N-terminal pelB signal peptide and C-terminal LPETG sortase motif, and a His$_6$-tag. The plasmids were transformed into E. coli WK6 cells and 1 l of TB-medium (100 µg/ml ampicillin) were inoculated with 25 ml of a LB-medium (100 µg/ml ampicillin) preculture. The culture was grown at 37 °C and 140 rpm until a cell density of 0.6 and induced with a final concentration of 1 mM IPTG in the medium. The protein expression was performed overnight at 30 °C and 140 rpm and the cells were harvested by centrifugation at 4000 × g for 20 min. The cell pellets were stored at −80 °C until further use.

For lysis of QM-VHHs, the pellet was resuspended in 15 ml extraction buffer containing 200 mM Tris (pH 8.0), 0.65 mM EDTA, and 500 mM sucrose and incubated for 1 h under gentle shaking at 4 °C. The solution was treated with 70 ml of 0.25 times diluted extraction buffer and incubated overnight at 4 °C under gentle shaking for osmotic lysis. Afterward, the solution was centrifuged at 8000 × g for 40 min and the supernatant was filtered through a 0.45 µm filter. For the HiSiaP-VHH, the cells were lysed by sonication (40% amplitude, 5 min, pulses 10 sec on-5 s off), centrifuged for 20 min at 20,000 × g and 4 °C, and also filtered through a 0.45 µm filter.

For both types of VHHs, the soluble fraction after lysis was filtered and mixed with Ni-NTA beads, previously equilibrated with 0.25 times diluted extraction buffer, and incubated for 1 h at 4 °C under gentle shaking. The suspension was loaded onto a bench-top column at room temperature, the flowthrough was discarded and the column was washed with 50 ml wash buffer (50 mM Tris (pH 7.5), 150 mM NaCl, 10 mM imidazole). Subsequently, the protein was eluted from the column with 10 ml elution buffer (50 mM Tris (pH 7.5), 150 mM NaCl, 500 mM imidazole) and concentrated to 5 ml with a Vivaspin (MWCO 3 kDa). The solution was loaded onto a HiLoad Superdex 75 16/600 on an ÄKTA chromatography system, equilibrated with 10 mM Tris (pH 7.3) and 140 mM NaCl for QM-VHHs and 50 mM Tris (pH 8) and 50 mM NaCl for the HiSiaP-VHH. The eluted fractions were checked on an SDS–PAGE, and protein-containing fractions were combined, concentrated, flash-frozen, and stored at −80 °C.

### Cloning, expression, and purification of a megabody (Mb3). The gene design and cloning procedure for a HopQ megabody was based on Uchański et al. [28]. The hopQ gene was ordered at Addgene and cloned into the pHEN6 vector to yield the same fused tags as the VHHs. Afterwards the VHH gene of VHH$_{QM}$3 (without the LPETG-sortase motif) was cloned into the megabody sequence via two SapI restriction sites, yielding the HopQ megabody (Mb3). The expression, lysis, and purification of Mb3 were performed as described for the VHHs. For the final purification step, a HiLoad Superdex 200 16/600 column was used.

### Site-specific biotinylation of HiSiaQM. For specific biotinylation of HiSiaQM, QuickChange mutagenesis after Liu et al. [71] was used to design a non-cysteine HiSiaQM construct (C94A, C325S, C334S, C400S, and C458S) in which cysteines (E235C or K273C) were introduced (Supplementary Table 4). Expression and purification were performed as described for the wild-type protein above. The labelling was performed during Ni-affinity chromatography of the HiSiaQM purification process. After 100 ml wash of the column with buffer B, supplemented with 22 mM imidazole, the column was washed with 50 ml buffer B with 1 mM TCEP, and again with 50 ml buffer B. The protein was eluted with 15 ml buffer B with 250 mM imidazole and supplemented with 100 times excess of a biotin-maleimide label (Maleimide-PEG2-biotin (EZ-Link$^{TM}$), Thermo Fisher Scientific). The solution was incubated overnight at 4 °C and further purified on a Superdex increase of 200 10/300 as described above.

### SPR binding characterization of VHH/megabody and HiSiaP to HiSiaQM. The SPR experiments were performed on a Biacore$^{TM}$ 8K instrument (GE Healthcare Life Sciences), using a streptavidin-functionalized sensor chip (Serie S Sensor SA, GE Healthcare Life Sciences) and HiSiaQM buffer B at 25 °C chip temperature. The two biotinylated HiSiaQM constructs, E235C-biotin and K273C-biotin, were injected and immobilized on the chip (100 nM, 5 µl/min, 70–100 s). The analytes, the VHHs or the megabody, were added in six injections (30 µl/min, 120 s) with a doubling of the concentration at each step, resulting in single-cycle kinetic titration curves. The binding data were double referenced by blank cycle and reference flow cell subtraction. For epitope binning, the VHHs were tested for competitive binding using an ABA-injection protocol.

For characterizing the binding of HiSiaP, biotinylated HiSiaQM (K273C) was immobilized as described above. Single-cycle kinetic titration curves were recorded for eight (0.078, 0.156, 0.312, 0.625, 1.25, 2.50, 5.00, 10.00 µM) injections of HiSiaP. The buffer was supplemented with 5 mM sialic acid where indicated. Competitive binding of VHH$_{QM}$3 and HiSiaP was assessed by consecutive injections of either buffer or VHH$_{QM}$3 (50 nM, 30 µl/min, 180 s) followed by HiSiaP (2.5 µM, 30 µl/min, 120 s) using a dual injection command.

All binding curves were fitted and analysed using the Biacore Insight Evaluation Software.

### ITC binding experiments. ITC experiments were performed according to Peter et al. [21] on a MicroCal PEAQ-ITC device from Malvern Panalytical, using corresponding MicroCalPEAQ-ITC software (version 1.21) for experiment design, measurement, and analysis. Sialic acid (Carbosynth) was dissolved in the P-domain standard buffer (50 mM Tris, 50 mM NaCl (pH 8)). Each mutant was at least measured three times and the mean values are given.

### Cryo-EM—sample preparation, data collection, processing, structure modelling. The cryo-EM structural studies were performed with HiSiaQM in MSP1D1-H5 nanodiscs with DMPC lipids. For the selection of HiSiaQM-containing nanodiscs, the purified Mb3 was loaded onto Ni-NTA beads, equilibrated with HiSiaQM buffer C, for 1 h at 4 °C. The flowthrough was discarded and the column was washed with 10 ml buffer C. The Mb3-bound Ni-NTA beads were resuspended in 2 ml buffer C, transferred into a flask, and supplemented with a HiSiaQM-nanodisc reconstitution mix after dialysis. The mixture was incubated for 1 h at 4 °C and transferred back to the bench-top column. The flowthrough was discarded, the column was washed with 10 ml buffer C and the protein was eluted with 1.5 ml buffer C with 500 mM imidazole. After concentration to around 50 µl in a Vivaspin (MWCO 100 kDa), the protein solution was loaded onto a Superose increase 6 3.2/300 column on an Agilent HPLC 1260 infinity II, previously equilibrated with buffer D (20 mM Tris (pH 7.5) and 100 mM NaCl). The eluted protein was monitored and fractionated manually and dropwise. The fractions that corresponded to the main protein peak were combined and used for cryo-EM experiments.

The grid preparation was performed on a Vitrobot (Thermo Fisher Scientific) at 4 °C and 100% humidity by using Quantifoil R1.2/1.3 grids. The blot time was set to 6 s and the blot force to 0. The grids were stored in liquid nitrogen until data

collection at a Titan Krios microscope (Table S1). A dataset of 5004 movies was collected using a Cs-corrected Titan Krios electron microscope (Thermo Fisher Scientific) operated at 300 kV, and equipped with a Gatan K3 camera and a BioQuantum imaging filter (Gatan). Images were recorded over 2.264 s with the camera operating in counting mode, with a dose rate of 22.179 e$^-$/Å$^2$/s for a total dose of 50.213 e/Å$^2$ over 50 frames. After patch motion correction and CTF-estimation with cryoSPARC[35], 2.33 million particles were automatically picked using a blob-picker job and subjected to multiple rounds of 2D classification. Representative 2D classes are shown in Fig. S3 and clearly show the megabody–HiSiaQM complex in the top views. The 2D classes and the movies were used as inputs for a template picker job resulting in 2.55 million particles that were subjected to one round of 2D classification. The remaining 2.29 million particles were exported from cryoSPARC using the csparc2star.py script by Daniel Asarnow (https://doi.org/10.5281/zenodo.3576630) and imported into RELION[36] to construct an ab initio model and to perform two rounds of 3D classifications. The first round (regularization parameter $T = 4$) revealed one good 3D class with a clearly visible transporter consisting of 790 K particles. A mask around the transporter was built in ChimeraX and was used as input for the second round of 3D classification without alignment ($T = 4$). Further 3D subclassing did not lead to classes with improved resolution and thus, the 215 K particles constituting the best class from the 2nd 3D classification were again imported into cryoSPARC for non-uniform refinement, and particle subtraction to remove the nanodisc density, and ultimately local NU refinement. The final map had a GSFSC resolution of 4.7 Å with a local resolution of the core area up to 3.7 Å. The map was subjected to local anisotropic sharpening in PHENIX[73]. The model was manually built in COOT[49] and refined with ISOLDE[74]. Temperature factors were refined with real-space refinement in PHENIX[73] and the geometric quality of the model was evaluated with MOLPROBITY[75].

**TRAP transporter in vivo growth assay.** The growth assay was performed in modified *E. coli* cells (SEVY3 cells) without their native sialic acid nanT (ΔnanT) transporter[54]. SEVY3 is a derivative of BW23115ΔnanT[8] carrying two further unmarked, in-frame deletions of the repressor genes, *nanR* and *nagC*, respectively, controlling the expression of the NanATEK-YhcH and NagAB branches of the sialic acid utilization pathway in *E. coli*[76–78]. This strain was obtained by sequential allelic replacement of the WT genes with "scar" sequences (coding for 19-residue-long "vestigial" peptides), made by SOEing PCR and cloned in the conditional counter-selectable vector, pKO3[79], which were then cycled through the parental strain using the method outlined in the same reference. SEVY3 was confirmed by PCR and sequencing of all three deletion loci.

A plasmid (pES7[8]) containing an ampicillin resistance and the genes for HiSiaP and HiSiaQM, was transformed into competent SEVY3 cells, as described before in Mulligan et al.[8]. If necessary, mutations were previously introduced via QuickChange mutagenesis[71] and checked via sequencing. For co-transformation and co-expression of VHHs in SEVY3 cells, the VHH genes were amplified with or without pelB signal sequence and cloned into a pET28a vector with kanamycin resistance. The VHH-containing plasmids were transformed into competent SEVY3 cells which contained the HiSiaPQM plasmid. For VHH-supplemented growth assays, the positive control with HiSiaPQM and without VHH was co-transformed with an empty pET28a vector. Expression levels of selected pET28a-VHH constructs (with and without the pelB signal sequence) were checked by Western blot using an anti-His primary antibody (6x-His Tag Monoclonal Antibody (4E3D10H2/E3) from Invitrogen/ThermoFisher; 1:1000 dilution) and m-IgGk BP-HRP (Santa Cruz; 1:5000) for detection.

For the TRAP transporter growth assay, the SEVY3 cells with the transformed plasmids were grown in 5 ml LB-medium with corresponding antibiotics at 37 °C and 140 rpm overnight. The cells were harvested at $4000 \times g$ for 15 min, the supernatant was discarded and the pellet was resuspended in 10 ml M9 minimal medium. The cell suspension was centrifuged under similar conditions and the washing procedure was repeated. After a third similar centrifugation step, the pellet was resuspended in 5 ml M9 minimal medium. 1 ml of this washed culture was used to inoculate 50 ml M9 minimal medium, supplemented with 1 mM IPTG, 2 mM MgSO$_4$, 0.1 mM CaCl$_2$, 0.1% (w/v) sialic acid (Carbosynth), and corresponding antibiotics. The cultures were grown for 17 h at 37 °C and 140 rpm in baffled flasks and the cell density was regularly measured on a NanoDrop 2000 (Thermo Fisher Scientific) at 600 nm.

**Site-specific fluorophore labelling of HiSiaP and VHH$_{QM}$3.** The cysteine mutants of HiSiaP (K254C) and VHH$_{QM}$3 (S85C) were created with QuickChange mutagenesis after Liu et al.[71]. All proteins were expressed and purified as described above in the corresponding chapter. Immediately before labelling, each protein solution was treated with reducing agent TCEP (Tris(2-carboxyethyl)phosphine) to a final concentration of 1 mM and incubated for 30 min on ice. Afterward, the TCEP was removed with a PD Miditrap G25 column (Cytiva), according to the manufacturers' instructions. The eluted protein fraction was directly incubated with a 5 times molar excess of AF-555 or AF-647 maleimide fluorophore (Jena-Bioscience) and incubated for 3 h at 4 °C. From this step on, illumination of the protein sample was avoided as much as possible. The protein solution was concentrated and washed with a Vivaspin (MWCO 3 kDa) to remove the most excess label. A size-exclusion chromatography (Superdex 75 10/300 on an ÄKTA system)

was performed to fully remove the unbound label and to check the successful labelling with the detection of the fluorophore-corresponding wavelength of the eluted protein solution. The protein-containing fractions were concentrated, flash-frozen, and stored at −80 °C.

**Solid supported bilayer preparation.** For the preparation of planar bilayers on glass supports, very small unilamellar vesicles (VSUVs) were prepared from detergent solution by the addition of cyclodextrin according to Grein et al. and Roder et al.[80,81].

For each bilayer, a lipid mixture with 31.8 mM DOPC (Avanti Polar Lipids, Birmingham, AL, USA) and 0.01 mol% TopFluor-PC (Avanti Polar Lipids) was prepared in chloroform, and the chloroform was slowly removed in a nitrogen stream. The resulting lipid film was solubilized in 200 µl HEPES buffer (20 mM HEPES, 150 mM NaCl, pH 7.4) supplemented with 40 mM Triton X-100. The lipid-detergent solution was split into 10 aliquots each 20 µl and stored at −4 °C until use. A second stock solution contained 4 mM heptakis(2,6-di-O-methyl)-β-cyclodextrin (cyclodextrin) in ddH$_2$O and was also stored at −4 °C. A suspension of VSUVs was prepared by first diluting the lipid-detergent stock solution in 200 µl HEPES buffer, followed by the addition of 1 µl HiSiaQM solution (HiSiaQM in DDM-micelles was first diluted 1:500 in HEPES buffer) and finally 200 µl of cyclodextrin stock solution and immediate, thorough mixing by vortexing for 2 min. Vesicles were generally used within 1 h of preparation.

Coverslips (18 × 18 mm) were cleaned overnight in fresh Piranha solution (one-part H$_2$O$_2$ 30% and two parts concentrated H$_2$SO$_4$), rinsed thoroughly with milliQ water and dried in a nitrogen stream. Clean coverslips were placed into custom-built sample chambers with an O-ring as a seal and two metal clips to fix the metal insert on top of the coverslip.

Bilayers were prepared immediately by adding 400 µl of freshly made vesicle suspension filling the well of the sample chamber. Due to electrostatic interaction between the lipid headgroups and the highly hydrophilic glass surface, vesicles are readily attached to the coverslip. The high surface tension led to the fusion of VSUVs and the formation of homogeneous bilayers on the complete cover slip within 5 min. Residual, non-fused vesicles were removed by carefully adding 2 ml of HEPES buffer to the sample chamber and then removing only 1 ml of liquid. In total, the sample was washed 12 times by adding 1 ml of HEPES buffer and removing 1 ml. The final volume of HEPES buffer in the chamber was 1.4 ml. During the washing steps, care was taken to not dry out the lipid bilayer.

**Bilayer binding assay and single-molecule imaging.** For the inhibition test and nanobody staining of HiSiaQM 10 µl Nb3 or 2 µl Nb3-AF-555 were added to the surface and incubated for at least 30 min. For Nb3-AF-555 the bilayer was washed 5 times afterward.

The P-domains variants were diluted 1:20 in HEPES buffer with 10 mM sialic acid, incubated for at least 20 min, and centrifuged at $14,000 \times g$ for 10 min. Samples without sialic acid were treated the same way. To each bilayer 1 µl of this solution was added. The buffer solution in the sample chamber was mixed carefully by pipetting up and down and incubated for 5 min before measurements were started.

Images were acquired at a custom-built, single-molecule sensitive, inverted microscope capable of total internal reflection fluorescence (TIRF) microscopy, which was equipped with an sCMOS camera (Prime BSI, Teledyne Photometrics, Tucson, AZ, USA)[81,82].

Illumination with total internal reflection reduced fluorescence excitation to a thin region at the coverslip surface with the benefit of background suppression from fluorescence outside the illuminated region. The illumination beam angle was adjusted by tilting a collimated laser beam in the object focal plane of the imaging lens until total reflection at the coverslip/medium interface was reached. This was accomplished by moving the laser beam focus laterally in the back focal plane of the objective, respectively, in a conjugated plane located outside the microscope. Using a ×63 objective lens with a NA 1.45 (Zeiss) resulted in a pixel size of 103 nm.

The focus was always carefully adjusted to the TopFluor-PC signal in the bilayer, which was excited by a laser emitting 405 nm (LDM-XT laser series, Lasos, Jena, Germany). The focus was stabilized during the measurements by the definite focus system (Zeiss).

For data acquisition, firstly 1000 frames using 640 or 561 nm laser excitation for visualizing the P-domains labeled by AF-647 or the Nb3-AF-555, respectively, and then 100 frames using 405 nm were acquired. The exposure time was set to 10 ms and only the central 200 × 200 pixels of the camera chip were read out. For each sample, 30 measurements were performed and each experiment was repeated for three independent samples.

Processing of the image sequences was performed in Fiji (version 1.52p)[83]. The 1000 frames of the P-domain were extracted. Next, the histogram was adjusted to minimum = 80 and maximum = 140, and the background was subtracted using a rolling ball radius of 10.

Tracking of single P-domains was performed using the Trackmate plug-in for ImageJ[84] (Supplementary Movie 4, Supplementary Fig. 19). For spot detection, the LoG-based detector was chosen. The parameter 'estimated blob diameter' was set to 0.75 µm and 'sub-pixel localisation' was activated. A threshold of 1 was used for spot filtering. For tracking the 'Simple LAP Tracker' was chosen. Gap closing was allowed with a maximum closing distance of 1 µm and a maximum frame gap of

two frames. The maximum linking distance was set to 1 µm. Each track is considered as a single interaction of the P-domain with the bilayer.

Three independently prepared samples were measured 30 times each, resulting in a total of 90 individual movies for each condition. Normalization of interactions per second was achieved by first dividing the total number of interactions by the summed total duration of all measured movies in the respective condition. Then, the positive control value was set to 1 and all other values were adjusted accordingly.

**Structural predictions with AlphaFold**. The source code of the AlphaFold[37] algorithm was downloaded from https://github.com/deepmind/alphafold and installed as described https://github.com/deepmind/alphafold. The pLDDTs scores were mapped onto the structures with PyMOL (www.pymol.org). The models of the tripartite complex are available as Supplementary Data 1 (outward facing) and Supplementary Data 2 (inward facing).

**Reporting summary**. Further information on research design is available in the Nature Research Reporting Summary linked to this article.

## Data availability
The coordinate and map data generated in this study have been deposited in the PDB and EMDB databases under accession codes 7QE5 and EMD-13930. The movie data generated in this study are provided as Supplementary Movies 1–4. The models of the outward and inward-facing tripartite complex are provided in Supplementary Data 1 and 2. Data underlying all plots are provided as Source data. The coordinate data used in this study are available in the PDB database under accession codes 5UL9, 2CEY, 3B50, 5NVA, 2HZL, 2ZZV). Source data are provided with this paper.

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

## Acknowledgements

This project was financed by the German Research Foundation (Deutsche Forschungsgemeinschaft, DFG) in project nos. HA 6805/4-1 and HA 6805/5-1 and a Method Development Grant by the TRA Life and Health (University of Bonn, Germany) as part of the Excellence Strategy of the federal and state governments (to G.H. and P.A.K.). M.F.P. acknowledges a Ph.D. fellowship from Konrad Adenauer Stiftung. U.K. received funding by the Deutsche Forschungsgemeinschaft (DFG)– Project-ID 398967434–TRR 261. M.G. is funded by the DFG under Germany's Excellence Strategy–EXC2151–390873048. We thank Elif Tokmak for technical assistance with the cloning of the HiSiaQM mutants. We thank Jan M.P. Tödtmann of the Core Facility Nanobody (CFN) at the Medical Faculty of the University of Bonn for cloning the nanobody library, selecting nanobodies by phage display, identification of hits, and their initial characterization by ELISA. We thank Florian I. Schmidt for establishing the CFN and infrastructure for camelid immunization and protocols for nanobody identification. We thank Sandy Macdonald (Biology Technology Facility, University of York, UK) for providing the hierarchical clustering tree of the VHHs. This work benefited from access to the Integrated Structural Biology platform of the Strasbourg Instruct-ERIC center IGBMC-CBI (Institut de génétique et de biologie moléculaire et cellulaire–Centre de Biologie Intégrative) Illkirch-Graffenstaden, France. We thank Nils Marechal and Corinne Crucifix of the IGBMC-CBI for their help with sample preparation and data collection. M.F.P. and G.H. thank the group of CZ (University of Regensburg, Germany) for initial cryo-EM experiments and the group of Bettina Böttcher (University of Würzburg, Germany) for preliminary cryo-EM data collections. G.H.T. and E.S. thank UKRI BBSRC for funding much initial work on this system & the BBSRC White Rose DTP for a studentship to S.T. and Phill Stansfeld (University of Warwick, UK) for building another initial model of SiaQM. M.F.P. and G.H. thank Janin Glaenzer and Olav Schiemann (University of Bonn, Germany) for support during the initial stages of this project. We are very grateful for critical feedback provided on a preprint (https://www.biorxiv.org/content/10.1101/2021.12.03.471092.abstract) of this article by the Biophysics Collab initiative by Rachelle Gaudet, Olga Boudker, Krishna Reddy and Valeria Kalienkova.

## Author contributions

G.H. acquired funding, planned, and supervised this study. M.F.P. prepared the protein samples, performed all experiments if not mentioned otherwise and contributed to each

experiment. P.D. and M.F.P. developed the reconstitution of the nanodiscs. P.-A.K. supervised alpaca immunization, selection, and initial characterization of VHHs. Proteins for the SPR experiments were prepared by M.F.P. K.G. and J.M. performed and analysed the SPR experiments, supervised by M.G. The VHH$_P$1 protein purification and ITC experiment was performed by N.S. Cryo-EM experiments were performed by M.F.P., G.H., and A.D. The cryo-EM dataset was processed by M.F.P. and G.H. The results were discussed with C.Z., V.H., and A.D. The TRAP transporter complementation assay was established and performed by M.F.P. with the help of E.S. who developed the original assay and k/o strain. S.T. provided the HiSiaQM multiple sequence alignments. M.F.P., J.P.S., and J.A.R. planned the single-molecule experiments, J.A.R. performed and analysed the single-molecule measurements; the results were discussed and evaluated with J.P.S. and U.K. J.A.R., J.P.S., and U.K. contributed to Fig. 6 and the manuscript. M.F.P. and G.H. wrote the manuscript together with G.H.T. All authors discussed the data and commented on the final manuscript version.

## Funding

## Competing interests

The authors declare no competing interests.
