## [Peer Review File · Nature Communications]

Structural and mechanistic analysis of a tripartite
ATP-independent periplasmic (TRAP) transporterREVIEWER COMMENTS

Reviewer #1 (Remarks to the Author):

In the manuscript by Peter et al., the authors describe the structural analysis of Tripartite ATP-independent periplasmic (TRAP) transporters, which are unique in their modular architecture where substrate transport is dependent on a periplasmic binding protein (P) as well as two transmembrane domains (QM). While structures of the P domain have been previously described, structures of the transmembrane domains have remained elusive. To facilitate structural analysis by cryo-EM, the authors develop a megabody to add 'bulk' to the 70-kDa transporter and can obtain a moderate resolution map of the QM domain. The development of heavy chain antibodies validates the orientation in the membrane, and the predicted tripartite complex and elevator mechanism is verified by a combination of computational models, mutational analysis, and genetic/biophysical assays. The structure of a monomeric elevator transporter, combined with the mechanistic validation, represents a significant advance in the field.

The first version of this manuscript was previously reviewed as a preprint by the Biophysics Colab initiative, of which I was part of the reviewing team. I applaud the authors for their serious consideration of the preprint critiques, which I believe is an essential gesture for facilitating an environment of open, accessible, and equitable science. The authors' detailed responses to these critiques, combined with the improved cryo-EM structure and the new single-molecule FRET data, have made for a compelling, well-designed, and rigorous manuscript with an elegant combination of structure, function, and mechanism. This manuscript will be of significant interest to researchers studying TRAP transporters and those generally interested in mechanisms of alternating access.

Essential revisions:

I do not understand the rationale for using a rooted phylogenetic tree to demonstrate the relationships of HiSiaQM-specific VHHs (Figure 3a), which implies temporal evolutionary events and that the VHHs 'descend' from VHHQM1. Furthermore, based on the information provided, it is unclear what the values at the tree nodes represent. Unless the authors can justify the usage of their current tree, a more informative figure to show the relative relationships of VHHs may be an unrooted maximum parsimony tree or a pairwise percent identity matrix, such as that integrated with Clustal Omega.

I congratulate the authors for the improvement of their cryo-EM structures since their preprint, which has significantly strengthened their manuscript. However, based on their described processing scheme, I wonder if additional major improvements in resolution may be possible, echoing my previous concerns from the Biophysics Colab review. This is based on 1) the large number of particles remaining from the

relatively non-specific cryoSPARC pickers, where single rounds of classification are often not sufficient - especially with so many particles and only four classes, and; 2) both 3D classification steps in Relion yielding classes containing ~1/3 of the total particle stack, which is indicative of possible 'junk' remaining in the stack. Specifically, I would like to see two approaches implemented – if the authors have already performed similar analyses, they should indicate as such in the text.

- o Continued 3D classification in Relion, where the selected class of 790,481 particles are put through another round of 3D classification. This should be continued iteratively until the reported resolution in Relion of the best class no longer improves.

- o Additional rounds of masked 3D classification w/o alignment, starting from the most recent stack containing 215,956 particles, until the reported resolution in Relion no longer improves.

- o After one (or both) of these approaches, the remaining stack may be much smaller but may contain higher-quality particles than the current best stack, possibly resulting in higher resolution structures after NU-refinement. The outcome of these exercises would be sufficient to convince me if further major improvements are possible.

If the map undergoes no significant improvements, side-chains should be 'stubbed' (under simple mutate in Coot, so that the sequence information is correct while avoiding ambiguous rotamer assignment) in the parts of the map where the side-chains are not well resolved prior to the release of PDB structures. This generally applies to regions outside the 'core' of the protein that do not achieve sub-4 Å resolution. After this, please fix the remaining rotamer/Ramachandran outliers and improve the clash score if possible.

I believe there was a misunderstanding in minor correction 8 regarding expression of VHHs in the complementation assay and a request for western blots. The basis of this request was that relative expression might change, even among variants of the same protein in the same expression system, which could result in different complementation results. Western blots would approximate relative cellular expression given equal total protein amounts. This is especially informative considering VHHQM2 and VHHQM5 would be expected to inhibit bacterial growth strongly yet do not appear to do so – lower expression could justify this discrepancy.

On a related note, I would like to see a similar western blot experiment to validate the relative expression of the mutants in Figure 5, assuming there is a tag on the protein that can be detected. Normalized SEC peaks and the intrinsic variability of protein purification do not eliminate the possibility that mutants are simply not as well expressed in cells, leading to decreased complementation. An orthogonal validation would be western blots of mutants in the plasmids/strains of the complementation assay – at the very least, the mutants that have decreased complementation compared to wild-type. I had recommended this previously; however, it did not make it into the Biophysics Colab review. Apologies for the oversight.

Figure 6: I would like to see more overall experimental detail in the methods and legend.

- o Based on the figure legend, it is unclear if the 'x' represents technical or biological replicates (the legend suggests that the data displayed is a single representative experiment). Please clarify.
- o Please include the total number of molecules analyzed, either directly in the figure or in a supplement. The methods indicate 30 samples, which I assume means movies because 30 molecules seem quite low for this type of experiment.
- o Please explain how the data was normalized to panel 6D.
- o Are the authors measuring multiple binding/unbinding events on a single molecule? Examples of raw traces in various conditions would be helpful. It would also be interesting to know how consistent the behavior is between molecules of the same sample, considering the bivalence of binding in SPR experiments, which could result in apparent binding heterogeneity in single-molecule experiments.

Figure 7: please include reproducibility statements for this figure and other SPR figures. Also, please include both K_D values from the bivalent analyte model for WT and E172R in the legend (currently, only the higher affinity value is displayed in the figure, and I cannot find the second affinity value for E172R in the text). Can E172R be well-fitted to a 1:1 binding model? If so, this mutant could be helpful in understanding the nature of bivalent P-QM binding.

Additional comments:

Line 144: The Crisman et al. paper (PNAS, 2009) describing the inverted topology of GltPh is now appropriately cited; however, only the instance of VcINDY is stated in the text. Please also mention GltPh in this context, as it adds to the validity and generalizability of the approach for elevator transporters.

Lines 191-192: I would also mention your single-molecule data here (regarding VHH competition for the periplasmic surface with P domain), which further strengthens the argument that VHH binds to the periplasmic face.

Line 202: Some preprints have systematically described and validated using flexible linkers in AlphaFold to predict complex formation (such as Ko & Lee, bioRxiv, 2021 - <https://doi.org/10.1101/2021.07.27.453972>), and subsequent studies have built upon this work. These should be cited.

Lines 283-285: While I appreciate the authors' thorough validation of mutants with SEC elution profiles and binding affinity to sialic acid to demonstrate the behavior of their proteins, upon taking a closer look, I do not necessarily agree with their statement regarding the similarity of QM proteins to WT. There appears to be variability of protein homogeneity following NiNTA affinity: in R30E, S356Y, and E429R, gels contain notable contaminating bands, and the first two mutants have apparent broader SEC peaks. These inconsistencies are potentially suggestive of differences in protein folding/stability. Thus, the statement should be toned down.

Line 335: Please define the bivalent analyte model further in the main text, which may not be apparent to readers unfamiliar with complex binding modes and SPR. Also, please clarify if the stoichiometry obtained from SPR excludes the possibility of an independent, lower-affinity binding site.

Figure 8: please indicate periplasmic vs. extracellular space.

If possible, I would encourage the authors to deposit VHH sequence(s) in Addgene if they so choose – the VHHs seem like handy tools that may benefit the greater scientific community.

I would also encourage including the tripartite complex PDB and a validation report in the supplemental material, which may be helpful to those wishing to perform MD simulations.

Reviewer: Krishna Reddy, Weill Cornell Medical College

Reviewer #2 (Remarks to the Author):

This manuscript describing the structure of a TRAP transporter, HiSiaQM, and modeling its interaction with a periplasmic substrate binding protein partner, SiaP, is much improved compared to the first version submitted to bioRxiv and evaluated by the Biophysics Colab.

Further processing has helped improve the cryo-EM map, improving the confidence in the model (also supported by AlphaFold predictions and mutagenesis). The authors have also added supported bilayer and TIRF experiments to determine that the periplasmic protein (SiaP) has highest affinity for the transporter when bound to sialic acid, and that its binding site on the transporter is indeed overlapping with the megabody used to determine the cryo-EM structure. Overall, this makes for a compelling manuscript describing the structure of TRAP transporters and a proposed mechanism.

A few issues should still be addressed:

(1) The title of paper should be either more specific (including HiSiaPQM) or at least switch to singular (“of a tripartite ATP-independent periplasmic (TRAP) transporter”) because the only transporter investigated in the manuscript is HiSiaPQM.

(2) The validation report for the structure (which was only a preliminary report, labeled as “not for manuscript review” by the PDB...) indicates many Ramachandran plot outliers – this should be resolved before the structure is published.

(3) In their response to the Colab report requesting that the expression of the VHHs be confirmed for Figure 3b, the authors state that they know the VHHs express because they use the same expression plasmid to produce the VHHs. But that is not what is described in the methods. Line 741 states “For protein expression, the genes were cloned into a pHEN6 vector...” and line 873-874 states “... cloned into a pET28a vector...”

Minor:

(1) The authors should carefully check the usage of commas throughout the manuscript – there are many inappropriately placed commas which could lead to misunderstandings.

(2) Line 120: grammatical issue “... domain is protrude to...”

(3) Line 126: “weak” is not an appropriate qualifier because the authors do not have a measurement of strength of this salt bridge interaction.

(4) Figure 3a: what is the difference between “x” and “n.d.”?

(5) Line 1100: “panel c-j” should be “panel d-j”.

Reviewer #3 (Remarks to the Author):

The TRAP transporter HiSiaPQM of *H. influenzae* is essential for virulence and host colonization. Understanding its structure/function at high resolution could provide the means to design antimicrobials for this pathogen which is now resistant to antibiotics.

TIRF microscopy, single particle observation and tracking are areas of my expertise. My evaluation of this manuscript will be short and focus only on the single particle data presented in Figure 6 as requested by the editor. My review of the single particle experiments is positive but I must leave the larger evaluation/determination of the paper's impact to the Cryo-EM expert(s) reviewing the data. However, as a non-expert in Cryo-EM and the associated computational methods, I was impressed with the level of structural detail that can be discerned for a transmembrane protein using these techniques and thought the data were well presented and could be appreciated by a novice.

The authors present the Cryo-EM structure of HiSiaQM and determine the orientation of the transporter using a megabody. HiSiaQM's structural similarity to the known VcINDY transporter suggests it functions to move substrate using an elevator mechanism. The authors developed an array of high affinity VHHs as determined by SPR and go on to show effective growth inhibition in a sialic acid transport assay when co-expressed with HiSiaPQM and periplasmically exported. A comprehensive model of the tripartite HiSiaPQM transport complex was validated through targeted mutagenesis and subsequent growth assays. The authors proceeded to observe and test assembly of the wild-type and mutant tripartite complexes using an in vitro single particle imaging assay in SSBs. Association of the wild-type P-domain to the QM-domains reconstituted into SSBs was demonstrated but, according to the authors, there was only a 50% reduction of P-QM binding events in the absence of sialic acid. A VHH effectively blocked the P-QM interaction as an essential control and mutations in the P-domain blocked or highly reduced association with the QM-domains. Finally, the authors employed SPR to show a sialic acid dependence of P-domain binding with immobilized QM-domains and binding was reduced using a VHH or mutant P-domains, consistent with the observations made in single particle SSB experiments.

Figure 6:

The single particle SSB experiments performed and analyzed in Figure 6 are straightforward and the methods employed to create SSBs are standard for the field. I believe the data make the point and are consistent with the general conclusions made by the authors. I have minimal requests to solidify the figure.

The Methods state 30 separate measurements (1000 images acquired for each measurement) were made for each sample in three independent experiments. A "track" was considered a single interaction, implying that the particles were motile. The authors show maximum projections for P-domain

interactions in 6d-j. Naively, the spots in a maximum projection should be blurred/larger than what looks like single point-spread-functions of static particles. However, particle dwell time on the SSB could be quite short (this is hard to gleem by reporting “normalized” interactions) and/or movement may be minimal but still should be evident if the lipid bilayer is fluid.

Reviewer request:

Please show a movie and/or tracks in a supplemental figure to show displacement over time to clarify.

The authors report a 50% reduction in P-QM binding events in the absence of sialic acid and were disappointed by this result. However, the lower panel of 6d (presence of sialic acid) when compared to 6e (absence of sialic acid) shows a much larger difference. The maximum projection ought to show at a glance the number of P-domain interactions over the 1000 frame acquisitions for one representative measurement. Simply counting the number of particles shown (~47 for 6d) and (~7 for 6e), there appears to be a large 85% (~7x) reduction in the absence of sialic acid in this example.

Reviewer request:

Please show another image for 6e that is representative of the mean reduction measured.

VHHQM3 nicely shows inhibition of P-domain binding. Although not a demand, but if the experiments were performed, please show the results of introducing a non-specific antibody in these single particle assays.

The results of using mutant P-domains lends excellent support to the authors interpretation of specificity.

REVIEWER COMMENTS

We would like to thank all referees for their time and effort in reviewing our manuscript.

Reviewer #1 (Remarks to the Author):

In the manuscript by Peter et al., the authors describe the structural analysis of Tripartite ATP-independent periplasmic (TRAP) transporters, which are unique in their modular architecture where substrate transport is dependent on a periplasmic binding protein (P) as well as two transmembrane domains (QM). While structures of the P domain have been previously described, structures of the transmembrane domains have remained elusive. To facilitate structural analysis by cryo-EM, the authors develop a megabody to add 'bulk' to the 70-kDa transporter and can obtain a moderate resolution map of the QM domain. The development of heavy chain antibodies validates the orientation in the membrane, and the predicted tripartite complex and elevator mechanism is verified by a combination of computational models, mutational analysis, and genetic/biophysical assays. The structure of a monomeric elevator transporter, combined with the mechanistic validation, represents a significant advance in the field.

The first version of this manuscript was previously reviewed as a preprint by the Biophysics Colab initiative, of which I was part of the reviewing team. I applaud the authors for their serious consideration of the preprint critiques, which I believe is an essential gesture for facilitating an environment of open, accessible, and equitable science. The authors' detailed responses to these critiques, combined with the improved cryo-EM structure and the new single-molecule FRET data, have made for a compelling, well-designed, and rigorous manuscript with an elegant combination of structure, function, and mechanism. This manuscript will be of significant interest to researchers studying TRAP transporters and those generally interested in mechanisms of alternating access.

Thank you for this positive evaluation of our work and your in-depth analysis of our manuscript!

Essential revisions:

I do not understand the rationale for using a rooted phylogenetic tree to demonstrate the relationships of HiSiaQM-specific VHHs (Figure 3a), which implies temporal evolutionary events and that the VHHs 'descend' from VHHQM1. Furthermore, based on the information provided, it is unclear what the values at the tree nodes represent. Unless the authors can justify the usage of their current tree, a more informative figure to show the relative relationships of VHHs may be an unrooted maximum parsimony tree or a pairwise percent identity matrix, such as that integrated with Clustal Omega.

The referee is of course right. We have changed the figure.

“Fig. 3 | Characterization of TRAP transporter specific VHHs and inhibition of transport *in vivo*. a, A hierarchical clustering tree of nine HiSiaQM specific VHHs, based on PBLAST e-values as the distance matrix for tree building. The binding affinities of the VHHs, determined from SPR experiments are given (x: no binding detected; n.d. not measured since no clear binding detected in size exclusion chromatography). VHHs that bind to HiSiaQM mutually exclusively are grouped by yellow and violet boxes. The underlying data are described in detail in Figure S9. HiSiaQM was immobilized on the SPR chip in two different orientations as indicated.”

I congratulate the authors for the improvement of their cryo-EM structures since their preprint, which has significantly strengthened their manuscript. However, based on their described processing scheme, I wonder if additional major improvements in resolution may be possible, echoing my previous concerns from the Biophysics Colab review. This is based on 1) the large number of particles remaining from the relatively non-specific cryoSPARC pickers, where single rounds of classification are often not sufficient - especially with so many particles and only four classes, and; 2) both 3D classification steps in Relion yielding classes containing ~1/3 of the total particle stack, which is indicative of possible ‘junk’ remaining in the stack. Specifically, I would like to see two approaches implemented – if the authors have already performed similar analyses, they should indicate as such in the text.

- o Continued 3D classification in Relion, where the selected class of 790,481 particles are put through another round of 3D classification. This should be continued iteratively until the reported resolution in Relion of the best class no longer improves.
- o Additional rounds of masked 3D classification w/o alignment, starting from the most recent stack containing 215,956 particles, until the reported resolution in Relion no longer improves.
- o After one (or both) of these approaches, the remaining stack may be much smaller but may contain higher-quality particles than the current best stack, possibly resulting in higher resolution structures after NU-refinement. The outcome of these exercises would be sufficient to convince me if further major improvements are possible.

We had indeed tried hard to further improve the reconstruction by performing further rounds of 3D classification in RELION, also with changing the regularization parameter T. However, visual inspection of the resulting 3D classes in ChimeraX did not show any further improvements. Also exporting these classes (down to 95K particles) to cryoSPARC and running NU refinement and local refinements did not improve the reconstruction. As requested, we have added a sentence to the Methods section that further 3D classifications did not improve the reconstruction:

“Further 3D subclassing did not lead to classes with improved resolution and thus, the 215 K particles constituting the best class from the 2nd 3D classification were again imported into cryoSPARC for non-uniform refinement, particle subtraction to remove the nanodisc density, and ultimately local NU refinement.”

We have tried our best to improve the resolution of the reconstruction and came to the conclusion that no further improvements are possible with the current dataset.

If the map undergoes no significant improvements, side-chains should be ‘stubbed’ (under simple mutate in Coot, so that the sequence information is correct while avoiding ambiguous rotamer assignment) in the parts of the map where the side-chains are not well resolved prior to the release of PDB structures. This generally applies to regions outside the ‘core’ of the protein that do not achieve sub-4 Å resolution.

We respectfully disagree on this point. We think that even at this moderate resolution, and even if not all atoms of a side chain are visible, including the side chain atoms is better than leaving the atoms out. After all, they must be there and even the approximate relative orientation of pairs of side chains can be very helpful for the interpretation of a structure. Furthermore, structures with stubbed sidechains can be misleading, especially when interpreted by non-experts. Also, parameters such as the clash score are affected by side-chain pruning.

After this, please fix the remaining rotamer/Ramachandran outliers and improve the clash score if possible.

We have done further refinements with ISOLDE and phenix.refine and this improved the overall stereochemical parameters of our structure significantly.

Here is the relevant excerpt from Table S1:

Model validation	
MolProbity ¹¹ score	1.2
Clash score	1.29
Rotamer outliers (%)	0.49
C-beta outliers (%)	0.73
Ramachandran plot	
Favored (%)	95.0
Outliers (%)	0.4

I believe there was a misunderstanding in minor correction 8 regarding expression of VHHs in the complementation assay and a request for western blots. The basis of this request was that relative expression might change, even among variants of the same protein in the same expression system, which could result in different complementation results. Western blots would approximate relative cellular expression given equal total protein amounts. This is especially informative considering VHHQM2 and VHHQM5 would be expected to inhibit bacterial growth strongly yet do not appear to do so – lower expression could justify this discrepancy.

Yes, this was a misunderstanding. Indeed, from our own experience and in accordance with reports from other labs (references Salema et al. and Pardon et al. below), the expression levels of different nanobodies can vary quite significantly, leading to the possibility that the failure of a strongly binding nanobody to inhibit transport (such as observed for VHH_{QM2, 5}) might simply be caused by low expression levels of that particular nanobody.

Therefore, as requested, we have checked the expression of the VHH_{QM2, 5} and 8 constructs (all of which had a low to no inhibitory effect) relative to the VHH_{QM7} construct (strongest inhibiting effect) with and without the PelB export signal by Western blotting. While there are slight differences in the expression levels, there is no obvious correlation between the binding affinity of the nanobody, its effect in the activity assay and its expression level. The Western Blot was added to the manuscript as Supplementary Figure 9c. We have also added a sentence to the main text, that an impact of the expression level on the inhibitory effect, or even on the non-inhibitory effect of VVH_{QM2} and 5, cannot be fully excluded.

- 1 Salema, V. & Fernández, L. Á. High yield purification of nanobodies from the periplasm of *E. coli* as fusions with the maltose binding protein. *Protein Expr Purif* **91**, 42-48 (2013)
- 2 Pardon, E. et al. A general protocol for the generation of Nanobodies for structural biology. *Nature protocols* **9**, 674-693 (2014).

“To exclude the possibility that the low inhibitory effect of VHH_{QMS} 2, 5, and 8 was merely due to much lower expression level of these VHH_{QMS} relative to the VHH_{QM7} construct (strongest inhibiting effect), we verified their expression by Western blotting (with and without the PelB export signal, Figure S9 cd). All VHHs were clearly expressed, but slight differences in the expression levels were indeed observed, which is a common finding for VHH expression^{57,58}. While there is no clear correlation between the individual expression level and inhibitory effects, we cannot exclude that the level of inhibition of the individual VHHs is to an extent biased by their expression level (Figure S9de).”

On a related note, I would like to see a similar western blot experiment to validate the relative expression of the mutants in Figure 5, assuming there is a tag on the protein that can be detected. Normalized SEC peaks and the intrinsic variability of protein purification do not eliminate the possibility that mutants are simply not as well expressed in cells, leading to decreased complementation. An orthogonal validation would be western blots of mutants in the plasmids/strains of the complementation assay – at the very least, the mutants that have decreased complementation compared to wild-type. I had recommended this previously; however, it did not make it into the Biophysics Colab review. Apologies for the oversight.

Since the proteins in the activity assay are indeed not tagged, we thought that testing all QM mutants with decreased complementation in the single molecule assay would be an alternative, orthogonal way to confirm the *in vivo* results. The match of the outcome of the

two techniques is very good indeed and the data have been added to the manuscript (see Figure 5, 6).

Figure 6: I would like to see more overall experimental detail in the methods and legend.

o Based on the figure legend, it is unclear if the 'x' represents technical or biological replicates (the legend suggests that the data displayed is a single representative experiment). Please clarify.

The "x" represents the average results of the three independently prepared samples, respectively. They indicate for every condition the reproducibility of the shown results. For each of these three samples, a fresh bilayer was prepared on a new coverslip and 30 individual measurements were performed. Thus, the "x" represents technical replicants for each condition. To clarify this issue, we have extended the figure legend accordingly.

o Please include the total number of molecules analyzed, either directly in the figure or in a supplement. The methods indicate 30 samples, which I assume means movies because 30 molecules seem quite low for this type of experiment.

For each of the three independently prepared samples, 30 measurements were performed. This means that each condition was measured altogether 90 times. A large number of molecules/ interactions were observed in the individual measurements. For example, a total of over 21,000 individual molecules/interactions were observed for the wild type HiSiaPQM with Neu5Ac with an average of ~200 interactions per measurement. The exact numbers are now included in Table S3.

o Please explain how the data was normalized to panel 6D.

Unfortunately, it is not clear to us, which exact figure was addressed here. In panel 6d, no normalization was performed. The upper image shows the first frame of an exemplary image sequence. The bottom image shows a maximum intensity projection of the corresponding image sequence. The maximum intensity projection converts multi-image sequences into single-image projections by selecting the pixel with the maximum intensity along the viewing direction (projection direction).

Presumably, the question referred to Figure 6k. Normalization was achieved by first dividing the total number of interactions by the summed total duration of all measured movies in the respective condition. Then, the value of the positive control was set to 1 and all other values were adjusted accordingly.

For clarification, we have added the following to the methods:

“Three independently prepared samples were measured 30 times each, resulting in a total of 90 individual movies for each condition. Normalization of interactions per second was achieved by first dividing the total number of interactions by the summed total duration of all measured movies in the respective condition. Then, the positive control value was set to 1 and all other values were adjusted accordingly.”

o Are the authors measuring multiple binding/unbinding events on a single molecule?

Examples of raw traces in various conditions would be helpful. It would also be interesting to know how consistent the behavior is between molecules of the same sample, considering the bivalence of binding in SPR experiments, which could result in apparent binding heterogeneity in single-molecule experiments.

As explained in the response to the first request of Reviewer #3, most interactions were immobile and short. However, longer mobile trajectories were occasionally observed in the bilayer. These could be caused by multiple interactions in sequence with different QM domains. Generally, the observation of several subsequent interactions of a single QM

domain with various P domains is unlikely. This is due to the fact that a great number of QM domains was present in the bilayer while the number of fluorescent P domains was low in order to enable observation of individual binding events. In this situation the encounter of several P domains with one QM domain in sequence is low. Therefore, conclusions in the suggested direction could not be made.

We have also created a Sup. Movie 4 that shows how the raw tracks compare to the raw data. The raw tracks for the data in Figure 6 are shown below and in Figure S19.

a-m, Maximum intensity projections (upper row) and raw traces (bottom row) of the measurements shown in Figure 6.

Figure 7: please include reproducibility statements for this figure and other SPR figures. Also, please include both kD values from the bivalent analyte model for WT and E172R in the legend (currently, only the higher affinity value is displayed in the figure, and I cannot find the second affinity value for E172R in the text). Can E172R be well-fitted to a 1:1 binding model? If so, this mutant could be helpful in understanding the nature of bivalent P-QM binding.

Based on the referees' suggestion, we performed multiple repeats of the SPR experiments with the tripartite complex. The interaction between the wildtype P-QM proteins was reproducibly observed in four independent experiments. For reasons discussed further below we have moved this data into the TIRF section of the manuscript.

The referees' comment concerning a possible different binding mode of the E172R mutant is a very good and thought-provoking point!

It prompted us to analyse this in more detail, including mutants from the other side of the interface, i. e. the QM domain. The resulting set of data was qualitatively consistent with the results in our manuscript, but, we found that the binding kinetics of the mutant SPR data were very difficult to interpret, because the introduced point mutants likely affected the different binding sites of the tripartite system to different extents. Also, with weaker affinities, the sensorgrams were much stronger affected by the unspecific binding of the P-domain to the chip surface. Taken together we found that this led to unstable fits, making it difficult to determine reliable K_{DS} over repeated measurements.

After careful consideration, we therefore decided to remove the mutant SPR data from the manuscript and instead plan to investigate the kinetics of the tripartite system in a follow-up study, including orthogonal methods to determine the mutant affinities. Clearly, this requires a lot of additional experimentation and is out of the scope of this manuscript.

However, because the wild-type SPR data so nicely backs up the TIRF data we decided to move this data into the TIRF section of the revised manuscript. For the reasons stated above, we also opted for a more qualitative interpretation of the wild-type binding data. We now simply state that binding is reproducibly observed by SPR, and that the mechanism is too complex to be described by a 1:1 model.

Regarding a reproducibility statement for the VHH SPR measurements, we now write in the legend of Figure S9 that the SPR data are based on single experiments. These high-quality measurements were primarily done to select the best binding VHHs for the other experiments in our study.

Additional comments:

Line 144: The Crisman et al. paper (PNAS, 2009) describing the inverted topology of GltPh is now appropriately cited; however, only the instance of VcINDY is stated in the text. Please also mention GltPh in this context, as it adds to the validity and generalizability of the approach for elevator transporters.

Done.

Lines 191-192: I would also mention your single-molecule data here (regarding VHH competition for the periplasmic surface with P domain), which further strengthens the argument that VHH binds to the periplasmic face.

Good point.

We added this line at the end of the paragraph:

“This conclusion is further supported by the single molecule data presented below.”

Line 202: Some preprints have systematically described and validated using flexible linkers in AlphaFold to predict complex formation (such as Ko & Lee, bioRxiv, 2021 - <https://doi.org/10.1101/2021.07.27.453972>), and subsequent studies have built upon this work. These should be cited.

Done.

Lines 283-285: While I appreciate the authors' thorough validation of mutants with SEC elution profiles and binding affinity to sialic acid to demonstrate the behavior of their proteins, upon taking a closer look, I do not necessarily agree with their statement regarding the similarity of QM proteins to WT. There appears to be variability of protein homogeneity following NiNTA affinity: in R30E, S356Y, and E429R, gels contain notable contaminating

bands, and the first two mutants have apparent broader SEC peaks. These inconsistencies are potentially suggestive of differences in protein folding/stability. Thus, the statement should be toned down.

As requested, we have toned the statement down:

“As mentioned above, the P-domain mutants had wild-type like affinity for sialic acid and all QM-mutants behaved overall similar to the wildtype protein in size exclusion chromatography experiments (Figure S18).”

Line 335: Please define the bivalent analyte model further in the main text, which may not be apparent to readers unfamiliar with complex binding modes and SPR. Also, please clarify if the stoichiometry obtained from SPR excludes the possibility of an independent, lower-affinity binding site.

As mentioned above, we decided to not use the bivalent analyte model and instead describe the remaining SPR data in a more qualitative fashion, so that the complex kinetics can be more thoroughly analysed in a follow up study.

Figure 8: please indicate periplasmic vs. extracellular space.

Done.

If possible, I would encourage the authors to deposit VHH sequence(s) in Addgene if they so choose – the VHHs seem like handy tools that may benefit the greater scientific community. The VHH sequences have now been added to the supplementary information (SI Figure 9c).

I would also encourage including the tripartite complex PDB and a validation report in the supplemental material, which may be helpful to those wishing to perform MD simulations. Agreed, we will upload the model as supplementary information. Of note, the usefulness of Validation reports for AF2 models have recently been critically discussed on the ccp4bb. Nevertheless, we will provide a molprobit summary report for the model.

Reviewer: Krishna Reddy, Weill Cornell Medical College

Reviewer #2 (Remarks to the Author):

This manuscript describing the structure of a TRAP transporter, HiSiaQM, and modeling its interaction with a periplasmic substrate binding protein partner, SiaP, is much improved compared to the first version submitted to bioRxiv and evaluated by the Biophysics Colab.

Further processing has helped improve the cryo-EM map, improving the confidence in the model (also supported by AlphaFold predictions and mutagenesis). The authors have also added supported bilayer and TIRF experiments to determine that the periplasmic protein (SiaP) has highest affinity for the transporter when bound to sialic acid, and that its binding site on the transporter is indeed overlapping with the megabody used to determine the cryo-EM structure. Overall, this makes for a compelling manuscript describing the structure of TRAP transporters and a proposed mechanism.

Thank you very much for your very positive overall assessment and your time and effort to review our work!

A few issues should still be addressed:

(1) The title of paper should be either more specific (including HiSiaPQM) or at least switch to singular (“of a tripartite ATP-independent periplasmic (TRAP) transporter”) because the only transporter investigated in the manuscript is HiSiaPQM.

Agreed:

“Structural and mechanistic analysis of a tripartite ATP-independent periplasmic (TRAP) transporter”

(2) The validation report for the structure (which was only a preliminary report, labeled as “not for manuscript review” by the PDB...) indicates many Ramachandran plot outliers – this should be resolved before the structure is published.

We re-refined the structure and improved the statistics.

Here is the relevant excerpt from Table S1:

Model validation

MolProbity ¹¹ score	1.2
Clash score	1.29
Rotamer outliers (%)	0.49
C-beta outliers (%)	0.73

Ramachandran plot

Favored (%)	95.0
Outliers (%)	0.4

(3) In their response to the Colab report requesting that the expression of the VHHs be confirmed for Figure 3b, the authors state that they know the VHHs express because they use the same expression plasmid to produce the VHHs. But that is not what is described in the methods. Line 741 states “For protein expression, the genes were cloned into a pHEN6 vector...” and line 873-874 states “... cloned into a pET28a vector...”

This was indeed a misunderstanding and has now been clarified, see Referee #1

Minor:

(1) The authors should carefully check the usage of commas throughout the manuscript – there are many inappropriately placed commas which could lead to misunderstandings.

Thank you. We have checked the commas.

(2) Line 120: grammatical issue “... domain is protrude to...”

Fixed.

(3) Line 126: “weak” is not an appropriate qualifier because the authors do not have a measurement of strength of this salt bridge interaction.

We have removed the qualifier.

“One of the few polar interactions in this interface is an ionic interaction between K45 and D242.”

(4) Figure 3a: what is the difference between “x” and “n.d.”?

x: no binding

n.d.: not determined.

This is now explained in the legend of Figure 3.

(5) Line 1100: “panel c-j” should be “panel d-j”.

Thanks for catching this. The figure has been extended and the panels should now be correctly cited.

Reviewer #3 (Remarks to the Author):

The TRAP transporter HiSiaPQM of *H. influenzae* is essential for virulence and host colonization. Understanding its structure/function at high resolution could provide the means to design antimicrobials for this pathogen which is now resistant to antibiotics.

TIRF microscopy, single particle observation and tracking are areas of my expertise. My evaluation of this manuscript will be short and focus only on the single particle data presented in Figure 6 as requested by the editor. My review of the single particle experiments is positive but I must leave the larger evaluation/determination of the paper's impact to the Cryo-EM expert(s) reviewing the data. However, as a non-expert in Cryo-EM and the associated computational methods, I was impressed with the level of structural detail that can be discerned for a transmembrane protein using these techniques and thought the data were well presented and could be appreciated by a novice.

Thank you!

The authors present the Cryo-EM structure of HiSiaQM and determine the orientation of the transporter using a megabody. HiSiaQM's structural similarity to the known VcINDY transporter suggests it functions to move substrate using an elevator mechanism. The authors developed an array of high affinity VHHs as determined by SPR and go on to show effective growth inhibition in a sialic acid transport assay when co-expressed with HiSiaPQM and periplasmically exported. A comprehensive model of the tripartite HiSiaPQM transport complex was validated through targeted mutagenesis and subsequent growth assays. The authors proceeded to observe and test assembly of the wild-type and mutant tripartite complexes using an in vitro single particle imaging assay in SSBs. Association of the wild-type P-domain to the QM-domains reconstituted into SSBs was demonstrated but, according to the authors, there was only a 50% reduction of P-QM binding events in the absence of sialic acid. A VHH

effectively blocked the P-QM interaction as an essential control and mutations in the P-domain blocked or highly reduced association with the QM-domains. Finally, the authors employed SPR to show a sialic acid dependence of P-domain binding with immobilized QM-domains and binding was reduced using a VHH or mutant P-domains, consistent with the observations made in single particle SSB experiments.

Figure 6:

The single particle SSB experiments performed and analyzed in Figure 6 are straightforward and the methods employed to create SSBs are standard for the field. I believe the data make the point and are consistent with the general conclusions made by the authors. I have minimal requests to solidify the figure.

We would like to thank the referee for taking the time to review the paper, and for acknowledging the potential of our work.

The Methods state 30 separate measurements (1000 images acquired for each measurement) were made for each sample in three independent experiments. A “track” was considered a single interaction, implying that the particles were motile. The authors show maximum

projections for P-domain interactions in 6d-j. Naively, the spots in a maximum projection should be blurred/larger than what looks like single point-spread-functions of static particles. However, particle dwell time on the SSB could be quite short (this is hard to gleem by reporting “normalized” interactions) and/or movement may be minimal but still should be evident if the lipid bilayer is fluid.

Reviewer request:

Please show a movie and/or tracks in a supplemental figure to show displacement over time to clarify.

In addition to the existing Movie S2, which shows the raw data, we have created Movie S3, which shows the raw data and the determined tracks in comparison. This presentation reveals that the measured signals are predominantly immobile.

In principle, we agree with the reviewer's expectation that the individual particles should be mobile. However, according to our experiences the mobility of proteins in solid supported bilayers often changes drastically compared to free standing bilayers due to the interaction of lipids and/or proteins with the coverslip surface. This resulted in an almost complete immobility of the QM domain in the SSBs (see Movie S1). We also tested the QM domain in free-standing bilayers, where we indeed observed the expected mobility and ensured functionality (data not shown). However, the SSBs were much easier to prepare, more robust and straight forward, which made them perfect for this interaction study.

Accordingly, the P domain is also predominantly immobile during the interaction with the QM domain. However, single mobile tracks can also be observed, which possibly show interactions of some P domains with multiple QM domains in sequence. However, the number of these events was very small compared to the immobile interactions.

As can be seen in Movie S2, most of these interactions were quite short. Occasionally longer interactions were observed. Overall, the residence times are negatively exponentially distributed, as is common for binding events in biological systems. The detailed distribution of the dwell times was not analyzed further in this work, because only a very small number of interactions were observed in most of the control measurements.

The authors report a 50% reduction in P-QM binding events in the absence of sialic acid and were disappointed by this result. However, the lower panel of 6d (presence of sialic acid) when compared to 6e (absence of sialic acid) shows a much larger difference. The maximum projection ought to show at a glance the number of P-domain interactions over the 1000 frame acquisitions for one representative measurement. Simply counting the number of particles shown (~47 for 6d) and (~7 for 6e), there appears to be a large 85% (~7x) reduction in the absence of sialic acid in this example.

Reviewer request:

Please show another image for 6e that is representative of the mean reduction measured.

Done.

VHHQM3 nicely shows inhibition of P-domain binding. Although not a demand, but if the experiments were performed, please show the results of introducing a non-specific antibody in these single particle assays.

Since all control experiments showed a very clear result, this experiment was not performed.

The results of using mutant P-domains lends excellent support to the authors interpretation of specificity.

Thank you!

** See Nature Portfolio's author and referees' website at www.nature.com/authors for

information about policies, services and author benefits.

This email has been sent through the Springer Nature Tracking System NY-610A-NPG&MTS

Confidentiality Statement:

This e-mail is confidential and subject to copyright. Any unauthorised use or disclosure of its contents is prohibited. If you have received this email in error please notify our Manuscript Tracking System Helpdesk team at <http://platformsupport.nature.com> .

Details of the confidentiality and pre-publicity policy may be found here

<http://www.nature.com/authors/policies/confidentiality.html>

Privacy Policy | Update Profile

DISCLAIMER: This e-mail is confidential and should not be used by anyone who is not the original intended recipient. If you have received this e-mail in error please inform the sender and delete it from your mailbox or any other storage mechanism. Springer Nature America, Inc. does not accept liability for any statements made which are clearly the sender's own and not expressly made on behalf of Springer Nature America, Inc. or one of their agents.

Please note that neither Springer Nature America, Inc. or any of its agents accept any responsibility for viruses that may be contained in this e-mail or its attachments and it is your responsibility to scan the e-mail and attachments (if any).

REVIEWERS' COMMENTS

Reviewer #1 (Remarks to the Author):

The authors have addressed most of my concerns. My one remaining concern is that the western blot in the new Supplementary Figure 9e is overloaded to the point where it prohibits meaningful analysis or interpretation. This experiment should be repeated with much less protein so that bands are distinguishable from one another. Also, the blot should include loading controls. I recommend this manuscript for publication, contingent on this correction.

I appreciate and accept the authors' justification regarding cryo-EM processing, and if they so desire, I recommend uploading the raw movies/micrographs to EMPIAR – perhaps future software will allow for improvements.

While I still disagree with the authors' decision to assign side chains in helices where only backbones can be reasonably assigned, many researchers do this, and this is a regular debate in the field with no consensus. The authors are sufficiently careful not to make serious conclusions regarding sidechain coordination and have not assigned Na⁺ ions in their structure (their evidence suggests but does not conclusively support similar Na⁺ sites as VcINDY). Therefore, this is acceptable.

Regarding the SPR experiments, our recent manuscript (Reddy et al, JGP 2022) described similar heterogeneous substrate binding in GlTPh, and our efforts to parse this out. Perhaps this would be useful to the authors as they explore this new direction.

Reviewer #3 (Remarks to the Author):

I recommend this manuscript for publication. The authors have address my concerns from the initial review.

REVIEWERS' COMMENTS

We thank all referees again for their time and effort to review our work!

Reviewer #1 (Remarks to the Author):

The authors have addressed most of my concerns. My one remaining concern is that the western blot in the new Supplementary Figure 9e is overloaded to the point where it prohibits meaningful analysis or interpretation. This experiment should be repeated with much less protein so that bands are distinguishable from one another. Also, the blot should include loading controls. I recommend this manuscript for publication, contingent on this correction.

We repeated the western blot with less sample loaded. We have added it to the Supplementary information. Our conclusions remain the same.

I appreciate and accept the authors' justification regarding cryo-EM processing, and if they so desire, I recommend uploading the raw movies/micrographs to EMPIAR – perhaps future software will allow for improvements.

Thank you!

While I still disagree with the authors' decision to assign side chains in helices where only backbones can be reasonably assigned, many researchers do this, and this is a regular debate in the field with no consensus. The authors are sufficiently careful not to make serious conclusions regarding sidechain coordination and have not assigned Na⁺ ions in their structure (their evidence suggests but does not conclusively support similar Na⁺ sites as VcINDY). Therefore, this is acceptable.

We agree that there are good arguments for both "camps" in the field and thank the referee to accept our decision to leave intact side chains in our structural model.

Regarding the SPR experiments, our recent manuscript (Reddy et al, JGP 2022) described similar heterogeneous substrate binding in GltPh, and our efforts to parse this out. Perhaps this would be useful to the authors as they explore this new direction.

Thank you. This is a very interesting study and we have added the reference to our manuscript.

Reviewer #3 (Remarks to the Author):

I recommend this manuscript for publication. The authors have address my concerns from the initial review.

Thank you!